

# The first magnetic observatory of Honduras: Assessment and magnetic prospecting in 2019 – 2025

Yvelice-Soraya Castillo-Rosales[1], Norman-Iván Palma-Cruz[1], Manuel-de-Jesús Rodríguez-Maradiaga[2,3], Félix-Enrique Rodríguez-García[2], Iván-Jorel Guerrero-Mejía[3], Christopher-William Turbitt[4], André-Jared Aguilar-Ochoa[1], Carlos-Alberto García-Osorio[1], Oscar-Rolando Mendieta-Brizuela[1], Isaías-Rafael Martínez-Hernández[1], Samuel-Elías Flores-Portillo[1], Jonathan-Luciano Vides-Zerón[1], Jean Rasson[5], John Riddick[7], Gerardo Cifuentes-Nava[6], Ana Caccavari-Garza[6], Natalia Gómez-Pérez[4]

[1]Department of Astronomy and Astrophysics, National Autonomous University of Honduras (UNAH), Tegucigalpa, 11101, Honduras
[2]Earth Sciences Honduran Institute (IHCIT), National Autonomous University of Honduras, 11101, Tegucigalpa, Honduras
[3]Department of Earth Sciences, National Autonomous University of Honduras, 11101, Tegucigalpa, Honduras
[4]British Geological Survey (BGS), Keyworth, Nottingham, NG12 5GG, UK
[5]Royal Meteorological Institute of Belgium (RMI), 3 Avenue Circulaire, B-1180 Brussels, Belgium
[6]Instituto de Geofísica, Universidad Nacional Autónoma de México (UNAM), Circuito de la Investigación Científica s/n, C.U., Coyoacán, 04150, CDMX, México
[7]Retired from the British Geological Survey.

*Correspondence to*: Yvelice-Soraya Castillo-Rosales (yvelice.castillo@unah.edu.hn), Norman-Iván Palma-Cruz (npalma@unah.edu.hn).

**Abstract.** Since 2017, the Departments of Astronomy and Astrophysics and Earth Physics at the National Autonomous University of Honduras have led efforts to establish the *First Magnetic Observatory of Honduras*, which is intended to provide continuous, precise, and permanent measurements of the Earth's magnetic field in the region. This initiative aims to supply critical data to the international scientific and commercial sectors. However, the project has faced significant challenges, including difficulties in locating a site that satisfies the strict requirements of magnetic observatories, bureaucratic inefficiencies, insufficient financial support, limited local engagement, and restrictions imposed by the COVID-19 pandemic. This paper highlights the importance of constructing a magnetic observatory in Honduras. It provides an in-depth analysis of four candidate sites: *La Tigra* National Park*, Francisco Morazán* (14º 13' 7.24", -87º 5' 16.87"); the First Communications Battalion in *Las Mesas, San Antonio de Oriente, Francisco Morazán* (14º 2' 23.42", -86º 56' 19.82"); the *Francisco Morazá*n Hydroelectric Power Station in *Cortés* (15° 02' 07", -87° 45' 04"); and the First Artillery Battalion in *Zambrano, Francisco Morazán* (14° 15' 09", -87° 25' 23"). The primary objective of this initial stage is to identify a vandalism-free site that meets the stringent magnetic cleanliness criteria required for the installation of an observatory. This process follows the guidelines of the *Manual for Magnetic Measurements and Observatory Practices* developed by the International Association of Geomagnetism and Aeronomy, along with expert recommendations from the INTERMAGNET Digital Geomagnetic Observatories network. The planned instrumentation will enable the observatory to (a) measure the Earth's natural magnetic field vector free from anthropogenic interference; (b) collect continuous, broadband, absolute, long-term time series data; and (c) monitor the local geomagnetic field and solar-geomagnetic activity on a continuous basis. To achieve these objectives, a



multidisciplinary team of faculty and students from the Faculties of Space Sciences and Sciences, in collaboration with experts from the British Geological Survey, has been assembled. Local personnel also gained valuable experience in magnetometry. Furthermore, offers of magnetometer equipment have been received from the Institute of Geophysics at the National Autonomous University of Mexico, Conrad Observatory in Austria, and INTERMAGNET Digital Geomagnetic Observatories network.

Keywords: Magnetic observatory, prospecting, magnetometer, magnetic gradient, geomagnetism.

## 1. Introduction

The Earth's magnetic field has a significant impact on technology, economy, navigation, and numerous other aspects of modern life. Measuring the magnetic field at the Earth's surface is crucial for understanding internal geophysical processes and their long-term variations. The speed of the north magnetic pole has increased from 0 to 15 km yr$^{-1}$ before 1990 to 50–60 km yr$^{-1}$ between 1990 and 2005. In October 2017, it crossed the International Date Line and began drifting southward toward Siberia (Livermore et al., 2020). This shift is significant because global navigation satellite systems(GNSS), ships, and aircraft rely on accurate magnetic models for navigation. Enhancing the precision of geomagnetic field measurements is essential for updating magnetic charts and models used in global navigation and directional geological drilling for oil and gas exploration (Reda et al., 2011). When the magnetic field changes, these models must be revised, as seen in the recent development of the World Magnetic Model High Resolution 2025, sponsored by the U.S. National Geospatial-Intelligence Agency (NGA) and the U.K.'s Defence Geographic Centre, and jointly developed by NOAA's National Centres for Environmental Information (NCEI) and the British Geological Survey (BGS).

Magnetic field measurements are also applied to studies of the Earth's crust, induced currents in power lines and railways, archaeological prospecting, mineral exploration, engineering investigations, oil drilling, navigation, and many other domains.

The term *space weather* refers to the effects produced by interactions between large amounts of radiation and electrically charged particles  from the Sun and beyond, acting on the near-Earth environment. The study of geomagnetism and space weather is crucial for preventing damage to health, technologies, and ecosystems, as significant disturbances of the Earth's magnetic field are caused by solar phenomena such as flares (powerful  eruptions carrying energetic protons, electrons, ions and radiation), coronal mass ejections (CMEs; millions of tons of solar plasma ejected at millions of degrees Celsius and millions of km s$^{-1}$), co-rotating interaction regions (where fast solar wind streams interact with slower ones), magnetic clouds (CMEs with highly energetic magnetic fields), and solar cosmic rays. When these events impact the Earth, they generate radiation storms, geomagnetic storms, radio blackouts, or solar energetic particle storms. These, in turn, produce electric currents that penetrate several kilometres deep into the Earth's mantle, affecting both biological and technological systems. Particularly dangerous are geomagnetically induced currents (GICs) in the Earth's crust and mantle, which can overload power



systems, induce currents in seabed cables and pipelines, and compromise critical infrastructure. At higher altitudes, severe space weather can disrupt HF and VLF communications for airlines and radio operators, increase ionising radiation exposure for astronauts and crews on polar flights, cause satellite failures, scintillation, or deorbiting, and damage GNSS navigation, satellite communications, power grids, and even trigger auroral activity.

Magnetic observatories are essential for studying space weather effects both at the ground and in space, as well as for forecasting and hazard mitigation. When severe space weather events occur, observatories detected and recorded them with magnetometers, helping to assess the risks that these electromagnetic currents pose for living beings, financial systems, power networks, satellites, GNSS, and all dependent technologies.

A magnetic observatory records precise, broadband, absolute, continuous, and long-term time series of the geomagnetic field (Borodin et al., 2011). The definitive data are periodically published for use by the scientific community as well as for practical and commercial applications.

The global network of magnetic observatories functions as a large-scale facility that is continually being expanded and improved. Achieving this requires the installation of more observatories, their even distribution across the planet, better instrumentation, reduced noise levels, improve temporal and amplitude precision, well trained personnel, continuity of data, robust data centres, and increased availability, accessibility, and quality of observations. The presence of dedicated local staff and suitable facilities is indispensable for establishing a successful observatory (Rasson et al., 2011).

The INTERMAGNET Digital Geomagnetic Observatories project (INDIGO) was created to foster the development of high-quality observatories in selected global locations. When equipment, software, training, or on-site data processing is lacking, INDIGO provides these resources so that local colleagues can begin or enhance their geomagnetic observations. A centralised website consolidates all current and past data from INDIGO observatories, including details of instruments in use, serial numbers, scale values, preliminary baselines, monthly bulletins, site plans, photographs, and historical records (Rasson et al., 2011). INDIGO specifically targets regions with gaps in global observatory coverage, such as Asia, Africa, and Latin America, thereby increasing the availability of data to the global scientific community (Borodin et al., 2011). INTERMAGNET (International Real-time Magnetic Observatory Network) certifies observatories worldwide for the quality of their data, with the highest density of observatories found in Europe. In contrast, Central America currently hosts only one operating observatory, located in Costa Rica.

Three types of instruments are required in a magnetic observatory: a variometer to record magnetic variations, a proton magnetometer to measure the absolute value of the field modulus and a declinometer/inclinometer (D/I instrument) to link the recorded variations with the absolute field components (Jankowski & Sucksdorff, 1996). The angle measured from true north at which the fluxgate magnetometer output is minimised is the declination angle. This measurement uses an azimuth mark



with a known direction relative to the true north. The true north may be determined from a reference target at least 100 meters

away or calculated via celestial navigation using the Sun or stars. Data from geomagnetic observatories must meet two criteria:

observations must form a continuous series, and they must be accurate and recorded in a magnetically clean environment

(Reda et al., 2011).

The infrastructure, equipment, and human resources (engineers, technicians, and researchers) required for an observatory are

costly, with equipment alone costing approximately 200,000 euros. For Honduras, the INDIGO project has offered a FLV1/A

triaxial fluxgate variometric magnetometer "LAMA", manufactured by the Royal Meteorological Institute of Belgium (RMI),

complete with dedicated data acquisition software. The D/I instrument has been quoted by both Mingeo and the RMI. This

consists on a fluxgate magnetometer mounted on a non-magnetic theodolite to perform absolute measurements of declination

and inclination. It enables compass calibration as well as periodic calibration of the variometers used in geomagnetic

observatories.

Accurate and long-term monitoring of geomagnetic variations requires careful calibration of variometer recordings. These

must be periodically referenced to absolute D/I measurements, with baseline variations interpolated using polynomials or

splines. Ensuring baseline stability  is a critical factor for evaluating an observatory's performance (Reda et al., 2011).

To guarantee a suitable magnetic environment for at least 50 years, observatories must be strategically located and designed

to minimise external disturbances. Modern magnetometers with high resolution and low drift, together with advanced

recording and processing systems, are essential for maintaining data integrity (Reda et al., 2011). The choice of site is

fundamental: it must be magnetically representative of its region, as determined through extensive surveys using prospecting

magnetometers and aeromagnetic maps. The site should exhibit minimal short-term and secular variations. A dense grid

survey, with measurement points spaced at $10 \times 10$ meters or less, is recommended to confirm field homogeneity. Sites with

large magnetic anomalies, particularly those exceeding several hundred nanoteslas, should be avoided (Jankowski &

Sucksdorff, 1996).

Artificial disturbances must be minimised, requiring observatories to be located at least 300 meters away from infrastructure

and several kilometres from electric railways. Electromagnetic interference, particularly from DC power lines, can seriously

compromise data reliability and must be considered during site selection and ongoing monitoring (Reda et al., 2011). Structural

and environmental factors also play crucial roles in ensuring measurement accuracy. Variometers should be installed on robust,

thermally stable, non-magnetic pillars, housed in insulated enclosures to minimise temperature-related measurement errors.

Since fluxgate magnetometers are sensitive to temperature fluctuations, protective measures such as reflective coatings and

thermal insulation are necessary (Jankowski & Sucksdorff, 1996).



The placement of sensors and electronic systems requires careful planning to avoid magnetic interference. Adequate spacing

between sensors --typically determined by the magnetic field of the instruments must be maintained. Electrical panels should

be positioned at least 15 meters from sensors, and all nearby materials must be verified as non-magnetic (Jankowski &

Sucksdorff, 1996). Construction materials such as concrete and metal fixtures must be carefully selected to avoid unintended

magnetic contamination (Krasnoperov et al., 2023).

Absolute magnetometers require fully non-magnetic pillars, ideally constructed from wood, limestone, or marble at the

measurement interface. While stability requirements are less stringent than for variometers, these pillars must still resist

environmental influences such as moisture and temperature changes. The location must provide a clear and unobstructed view

of an azimuth mark, ideally situated hundreds of meters away, to ensure precise directional alignment. The observatory must

remain accessible to scientific staff while being protected against vandalism and unauthorised interference. Establishment and

maintaining a geomagnetic observatory therefore demand rigorous site selection, structural design, and technological

adaptation to guarantee long-term reliability and accuracy of data.

Nevertheless, as Rasson et al., 2011 note, "the experience has shown that it is mainly the efforts of a small group of motivated

individuals, working within scientific institutions, with or without money for projects, who manage to make things progress,"

and Honduras has been no exception. The search for a site for the Honduran magnetic observatory began in 2019 and continues,

as much of the country's soil contains ferromagnetic materials. In addition, because vandalism is common, in 2022, the team

proposed locating the observatory within a Honduran Army battalion. In 2024 the *Francisco Morazán* Hydroelectric Power

Station in Cortés was also visited, as the soil there is predominantly limestone, but the site was ultimately deemed unsafe.

Contacts have been established at international workshops in Mexico and Brazil, with colleagues responsible for observatories

across the Americas, Africa, Oceania, Asia, and Europe, many of whom have offered technical and scientific support. Training

has been provided by the British Geological Survey (BGS), the Geophysical and Astronomical Observatory and the Research

Centre for Earth and Space of the University of Coimbra (OGAUC and CITEUC), the National Observatory of Brazil (ONB),

the Institute of Geophysics of UNAM, the Pan American Institute of Geography and History, and the Santa Elena Observatory

of the Costa Rican Electricity Institute (ICE). Additional advice has been received from Dr. Jean L. Rasson, John Riddick,

Christopher Turbitt (former and current administrators of BGS geomagnetic observatories), Natalia Gómez-Pérez (BGS) and

Thomas Martyn (BGS), Esteban Hernández (IG-UNAM, RIP), Gerardo Cifuentes (IG-UNAM) and Ana Caccavari (IPGH-

UNAM).

To date, the Magnetic Observatory of Honduras (MAGHO) team has conducted 45 days of fieldwork at 19 sites across the

country and carried out prospecting at five of them. Table 1 lists the sites, and Figure 1 shows their distribution on the map of

Honduras. Sites that were assessed but not selected for prospecting exhibited anthropogenic noise, visible ferromagnetic



materials, nearby power lines, difficult access, or excessive distance from headquarters. Other potential sites were  not visited

for similar reasons.

**Table 1: List of assessed sites from 2019 to 2025**

| # | Site name | Geographic coordinates | Visiting dates | Activities |
|---|-----------|------------------------|----------------|------------|
| 1 | The COPECO headquarters, *Danlí city, El Paraíso province* | 14.032°, -86.580º | 2019/05/10 | Assessment |
| 2 | The UNAH-El Paraíso campus, *Danlí city, El Paraíso* | 13.996°, -86.572° | 2019/05/10 | Assessment |
| 3 | The National Pedagogical University's (UPN) campus in *Danlí city, El Paraíso* | 14.007°, -86.572° | 2019/05/10 | Assessment |
| 4 | The Institute of Forestal Conservation, *Danlí city, El Paraíso* | 14.027°, -86.582° | 2019/05/10 | Assessment |
| 5 | *The El Picacho* National Park, *Tegucigalpa city, Francisco Morazán province* | 14.120°, -87.194° | 2019/06/28 | Assessment |
| 6 | *The El Piligüín* Park, *Tegucigalpa, FM* | 14.157°, -87.140° | 2019/06/28 | Assessment |
| 7 | *The Peña Blanca* mine, *La Tigra* National Park, *Francisco Morazán* | 14.213°, -87.090° | 2019/07/22 | Mag profiles |
| | | | 2019/09/27 | Mag mesh |
| 8 | *The Mirador* mine, *La Tigra* National Park, *Francisco Morazán* | 14.218°, -87.090° | 2019/08/23 | Mag profiles |
| | | | 2019/10/07 | Mag profiles |
| | | | 2019/10/17 | Topography |
| | | | 2019/10/18 | Topography |
| | | | 2019/11/08 | Topography |
| | | | 2019/11/13 | Mag mesh |
| 9 | *The La Tigra* Park surroundings, *Tegucigalpa, FM* | 14.228°, -87.090° | 2020/01/23 | Assessment |
| 10 | The Honduran Army's Defence College, *Tegucigalpa, FM* | 14.059°, -87.270° | 2020/02/27 | Assessment |
| | | | 2022/04/16 | Assessment |
| 11 | Ciudad Universitaria, *Tegucigalpa, FM* | 14.080°, -87.150° | 2021/04/16 | Assessment |
| 12 | The UNAH-*Choluteca* campus, *Choluteca city, Choluteca* province | 13.326°, -87.140° | 2021/05/28 | Assessment |
| 13 | The UNAH's Centre for Aquaculture and Fisheries Research (CIAP), *Choluteca, Choluteca* | 13.418º, -87.430º | 2021/05/28 | Assessment |
| | | | 2021/07/30 | Aerial prospecting |
| 14 | The First Communications Battalion at *Las Mesas, San Antonio de Oriente, Francisco Morazán* | 14.035°, -86.94° | 2022/04/16 | Mag mesh |
| | | | 2022/04/26 | Topography |
| | | | 2022/06/26 | Aerial prosp. |
| | | | 2022/06/27 | Aerial prosp. |
| | | | 2024/08/08 | Mag mesh |
| | | | 2024/08/23 | Mag mesh |
| | | | 2024/09/10 | Mag mesh |
| | | | 2024/09/11 | Mag mesh |



| | | | 2024/09/12 | Mag profiles |
|---|---|---|---|---|
| | | | 2024/09/13 | Mag profiles |
| | | | 2025/05/30 | Mag profiles |
| | | | 2025/08/08 | Mag profiles |
| 15 | The UNAH-*Atlántida* campus, *Atlántida* province | 15.737°, -86.86° | 2023/09/13 | Assessment |
| 16 | *The Francisco Morazán* Hydroelectric Power Station, *Cortés* province | 15.035°, -87.756 | 2024/09/26 | Mag. profiles |
| | | | 2024/09/27 | Mag. profiles |
| 17 | The Second Infantry Battalion at *Támara town, Francisco Morazán* province | 14.005º, -87.016º | 2024/10/10 | Assessment |
| 18 | The First Engineers Battalion at *Siguatepeque city, Comayagua* province | 14.188°, -87.333° | 2024/10/11 | Assessment |
| 19 | The First Artillery Battalion at *Zambrano, Francisco Morazán* province | 14.006º, -87.006 | 2024/10/10 | Mag profiles |
| | | | 2024/10/11 | Mag profiles |
| | | | 2024/10/24 | Mag profiles |
| | | | 2025/03/14 | Mag profiles |
| | | | 2025/04/04 | Mag profiles |
| | | | 2025/04/11 | Mag profiles |
| | | | 2025/04/25 | Mag mesh |

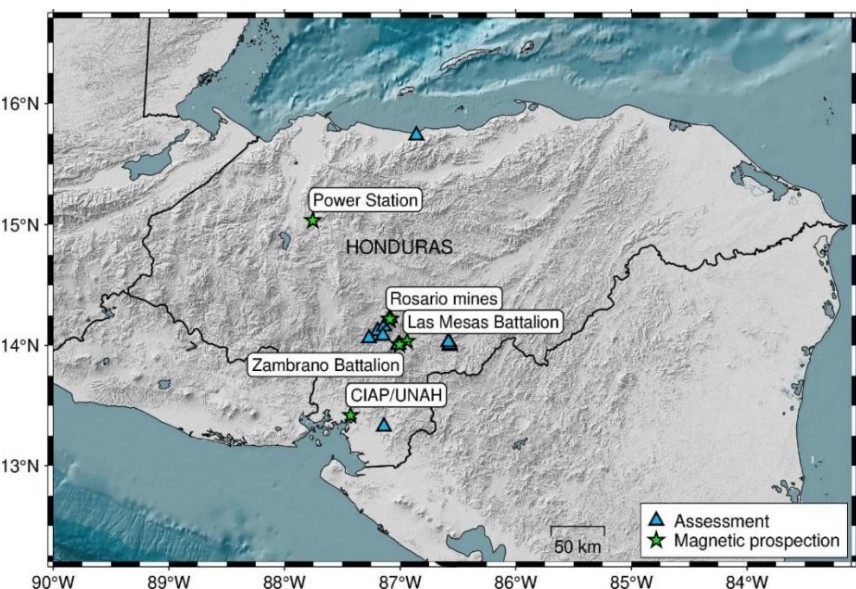

**Figure 1. Geographical distribution of the sites visited. Stars show the sites where prospecting was conducted, while triangles represent the remaining sites. Credits: Félix Rodríguez. Map generated with Python software©.**





## 2. Methodology

The main objective of this stage is to identify a site that meets the International Association of Geomagnetism and Aeronomy (IAGA) criteria for magnetic cleanliness for the installation of an observatory (see Jankowski & Sucksdorff, 1996, Newitt et al., 1997, Ministerio-de-Defensa, 2013,  Krasnoperov et al., 2023; Borodin et al., 2013). The key analysis categories are as follows:

- Magnetic isolation: The magnetometers must be located at least 380 m away from streets, power lines, buildings, motors, and traffic, as well as from people carrying telephones, weapons, or other metallic objects.

- Horizontal and vertical magnetic gradients: These must not exceed ~1 nT/m, measured with magnetometers and analysed with 2D and 3D plots.

- Safety

- Accessibility and proximity

- Environmental factors: Including temperature, moisture levels, non-ferromagnetic soil, vegetation, and related aspects.

As an initial step, we arranged meetings with key individuals and institutions to establish alliances,  cooperation agreements, donations, financial support, training opportunities, and to identify potential candidate sites. Stakeholders include UNAH rectors, directors of various UNAH campuses, the Honduran Institute of Earth Sciences (IHCIT), the UNAH Department of Earth Physics, the National Army, the National Electric Energy Company (ENEE), and the global observatory community (e.g., INDIGO, Santa Helena Observatory). Other organisations consulted included the Federal Institute for Geosciences and Natural Resources (BGR) of Germany, Geology and Geophysics Prospecting Company (PROGEO), the National Centre for Atmospheric, Oceanographic, and Seismic Studies (CENAOS), as well as national parks and foundations such as the *d* Foundation.

Following these meetings, the MAGHO team analysed potential candidate sites using Google Earth© and Google Maps©. Once initially suitability was discussed, field visits were conducted to evaluate anthropogenic noise, artificial magnetic field sources, accessibility, isolation, proximity to headquarters, soil composition, and environmental conditions.

At several sites, aerial surveys were performed using a MagDrone© SENSYS© magnetometer, loaned by the BGR of Germany, and a Matrice 600 Pro drone, loaned by the Small Central Units of the National Electric Energy Company (ENEE), within the framework of the *Yacimientos* II Project (*Yacimientos* meaning *reservoirs*), implemented by BGR and ENEE.

Most ground surveys were carried out using a portable Overhauser gradiometer GEM GSM-19G with a double coil, provided by PROGEO, and a portable Overhauser magnetometer GEM GSM-19P, provided by UNAH's Department of Earth Physics. Both instruments were manufactured by GEM Systems©. The gradient method eliminates the need for diurnal corrections and enhances the detection of environmental anomaly boundaries.

Survey outlines or polygons were established in north-south (N–S) or east-west (E–W) orientations, using a compass, 50 m measuring tapes, stakes, beads, and spray paint. For profile surveys, readings were taken every 2 m, while for mesh surveys,



2 m × 2 m grids were laid out. In practice, maintaining uniform grids was challenging due to topographic relief and forest
density.

The GEM GSM-19G coils were mounted on a stake, separated by 56 cm, with the lower sensor positioned 100 cm above the
ground. The GPS unit was placed 50 cm above the upper coil, connected to the recording and memory storage system. Field
notes were documented in a paper notebook to track data variability. Readings were recorded in the instrument's memory for
2 – 5 seconds. When conditions permitted, guide tapes were used to maintain alignment, as determined with a compass. In
other cases, a pre-measured 2 m stake was employed as a reference for measurements and alignment based on previously
marked points in the terrain.

The recorded data were later downloaded as ".txt" files and processed to generate 2D magnetic field intensity profiles, contour
plots, and bubble plots using Python, Matlab©, and Oasis Montaj©.

Some topographic surveys were carried out using two total stations:

- Trimble© Total Station (with a Sokkisha© tripod), provided by UNAH's Civil Engineering Laboratories, along with
accessories such as two surveying prisms, two CST Berger batons, a bipod, a compass, a 30 m tape measure, a 5 m
tape measure, two plummets, a 2 kg sledgehammer, two machetes, and stakes.
- NTS-362 SOUTH total station provided by the Department of Science and Technology of Geographic Information
(DCTIG) of the Faculty of Space Sciences, with an angular accuracy of ±2" (vertical and horizontal) and a distance
meter (2 mm + 2 ppm × D) with one prism.

Decisions regarding the viability of each site were based on the analysis categories listed above, as well as the plotted results.
A comparative assessment of data collected from different sites enabled us to determine the most favourable location for the
observatory.

**3.  Results and Discussion**

**3.1.  *The Danlí city, El Paraíso* province (13.99° N, 86.57° W)**

The first site assessment was conducted in May 2019, by Yvelice Castillo, accompanied by a CENAOS technician and Gerardo
López, head of the Permanent Contingency Committee (COPECO) in the city of *Danlí* , province of *El Paraíso*. The facilities
of COPECO in *Danlí*, the Institute of Forest Conservation, the National Pedagogical University *Francisco Morazán* campus,
and the UNAH-El Paraíso campus were visited. These sites were discarded because their available areas were smaller than the
required minimum and because major roads were located less than 300 m away.



### 3.2. The *El Picacho* (14.12°, -87.19°) and the *El Piligüín* (14.157°, -87.14°) Parks in *Tegucigalpa, Francisco Morazán*

On 28 June 2019, Yvelice Castillo, Norman Palma, and Carlos Luis Barahona visited *El Picacho* National Park and *El Piligüín* Park owned by the UNAH Workers' Union, in Tegucigalpa, Francisco Morazán province. Both sites were discarded due to anthropogenic noise and their proximity to major roads.

### 3.3. The *La Tigra* Park, *Francisco Morazán* province (14.22°, -87.082°)

In July 2019, Yvelice Castillo, Norman Palma, Manuel Rodríguez (Department of Earth Physics), and Jorge Murillo (Technical Director of the *Amitigra* Foundation, responsible for *La Tigra* National Park) inspected the old mining town of *Rosario* and located the *Peña Blanca* Mine. Although the site initially appeared suitable due to its remote setting and low cultural noise, several issues were identified.

### 3.3.1. The *Peña Blanca* Mine, *La Tigra* Park, *Francisco Morazán* province (14.2128°, -87.0939°)

The pithead was found to contain reinforced concrete, and the foundations of an old mechanical workshop with metallic remains were identified about 20 m inside. That day, a magnetic profile was conducted along the mine axis, and a magnetic grid was mapped at the entrance using the Overhauser magnetometer GEM GSM-19W with a single coil. Figure 2 shows a superposition of both surveys.

### 3.3.2. The *Mirador* Mine, *La Tigra* Park, *Francisco Morazán* province (14. 21779º, -87.08797º, 1775 masl)

The second mine evaluated is located near the *Mirador* site (which means "balcony") in *La Tigra* National Park. Figure 4 presents a magnetic contour plot from data measured on 23 August 2019 using the GEM GSM-19W magnetometer, operated by Heydi Martínez (IHCIT). The elevation contours measured are also shown. A clear correlation exists between elevation and the magnetic field gradient in a relatively homogeneous magnetic medium (Eppelbaum & Khesin, 2012): maxima correspond to ridges of the "magnetic relief", while minima correspond to valleys. However, the presence of a steel balcony disturbed the magnetic field (see left of Figure 4).

Because the GPS signal was lost inside the mine, UNAH's Civil Engineering Laboratories provided a Trimble total station to model the mine axis spatially. Engineer Maryuri García operated the instrument, assisted by Yvelice Castillo, Norman Palma, and engineer Carlos Luis Barahona (Department of Astronomy and Astrophysics). Some mine sections were extremely narrow, forcing the team to crawl while carrying the total station. The internal mine temperature ranged from 13 to 15 ºC. Table 2 summarises the magnetic surveys conducted between 23 August and 13 November 2019.



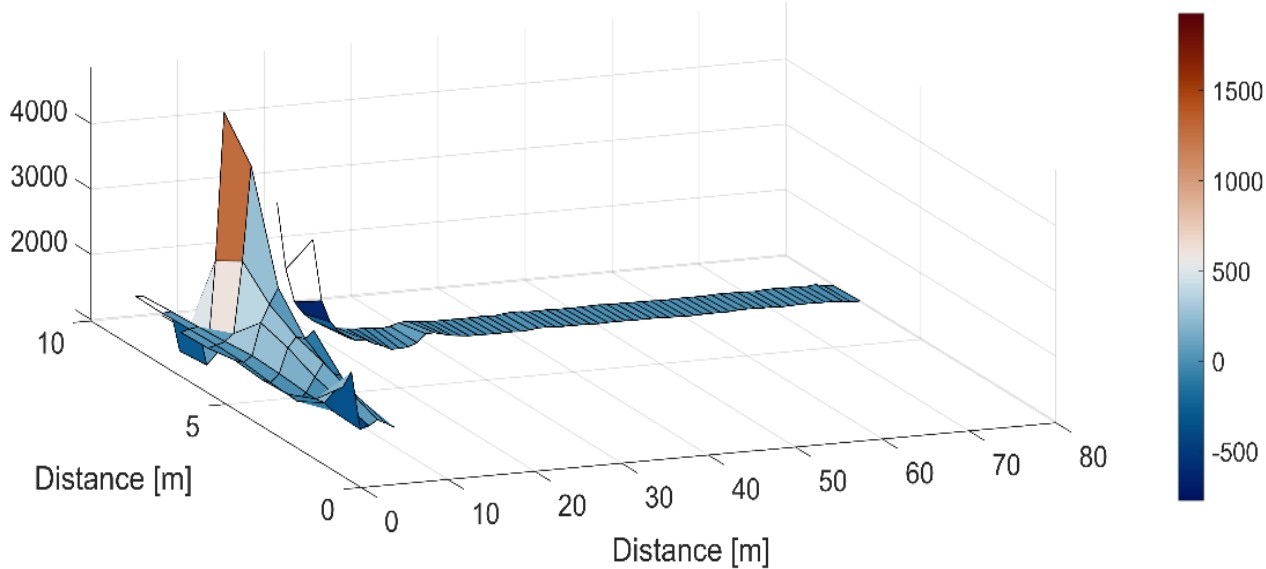

**Figure 2. Magnetic field intensity [nT] measured on a 35000 nT background in the *Peña Blanca* mine. The mesh on the left measures approximately 10 × 6 meters, with points spaced ~1 meter apart. The mine's pithead is located between this mesh and the profile on the right. The profile points are spaced ~1 meter apart along the mine axis. Plot generated with Matlab© software. Credits: Yvelice Castillo.**

On 8 November 2019, with the support of engineer Maryury Garcia, a topographic survey was performed using a total station
to connect points inside the mine with reference points on the mountain above. Figure 3 presents a top view of these surveys,
distinguishing between internal and external points.

On 13 November 2019, with the support of Félix Rodríguez (IHCIT), a 2 m step profile was measured along the *Mirador* mine
axis (interior) and a 11 m × 8 m mesh with 1 m spacing was surveyed on the mountain above point 6 (Figure 3). The UNAH's
GEM GSM-19W was used for this prospecting. Tapes, rods, stakes, red paint, sledgehammers, and machetes were used to
mark the grid points (see Figure 5).

The *Mirador* mine was ultimately discarded for the following reasons:

● the land does not belong to the Amitigra Foundation, so construction would depend on private landowners;

● security at *La Tigra* National Park is insufficient.

● humidity levels inside the mine were too high for equipment;

● significant anomalies were measured both in the mines and on the mountain.




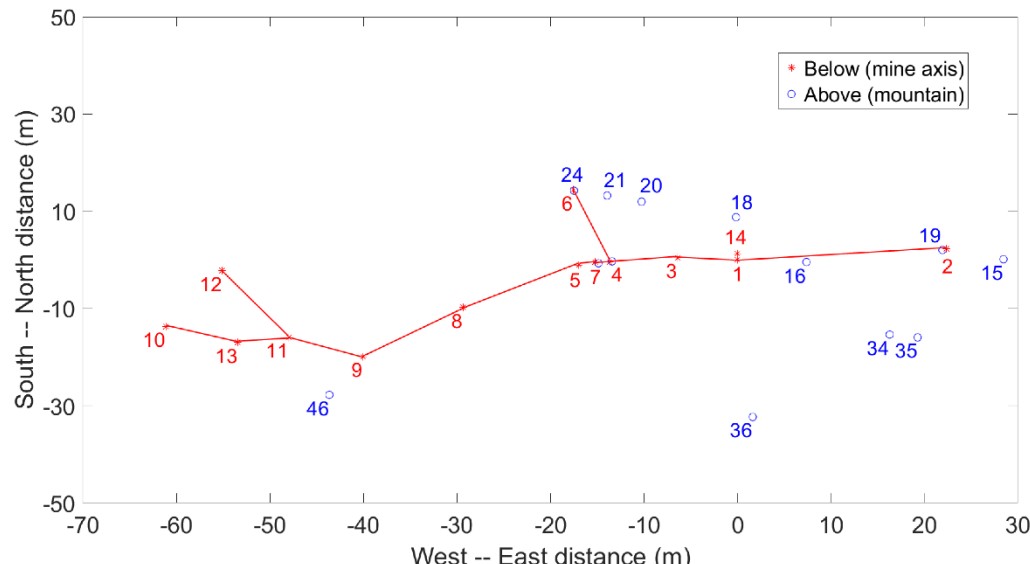

**Figure 3. Top view of the spatial distribution of the *Mirador* mine axis and points plotted with the total station on October 17, 2019. The red points correspond to the mine's interior. Point number 1 (on the right) is the mine's pithead. The blue points were taken on the mountain, above the mine. Credits: Maryuri García, Yvelice Castillo. Plot generated with Matlab©.**

**Table 2. Prospecting core information at *La Tigra Park, FM*.**

| # | Date | Site name | File name | Configuration and step | Number of points |
|---|------|-----------|-----------|------------------------|------------------|
| 1 | 2019/07/22 | *Mirador* to *Peña Blanca* pathway (exterior) | 07ros01 | 2 m, profile | 48 |
| 2 | 2019/07/22 | *Peña Blanca* mine axis (interior) | 08ros02 | 1 m, profile | 71 |
| 3 | 2019/08/23 | Area over the Mirador mine | 09ros03 | 1 m, mesh | 63 |
| 4 | 2019/08/23 | *Mirador* mine's pathway from point 1 to point 10 (see Figure 3) | 10ros04 | 5 m, profile | 20 |
| 5 | 2019/08/23 | *Mirador* mine's pathway from point 1 to point 6 | 11ros05 | 5 m, profile | 20 |
| 6 | 2019/11/13 | *Mirador* mine's pathway from point 15 to point 12 | 12ros06 | 2 m, profile | 68 |
| 7 | 2019/11/13 | Mountain over *Mirador* mine's point 6 | 13ros07 | 1 m, mesh | 131 |





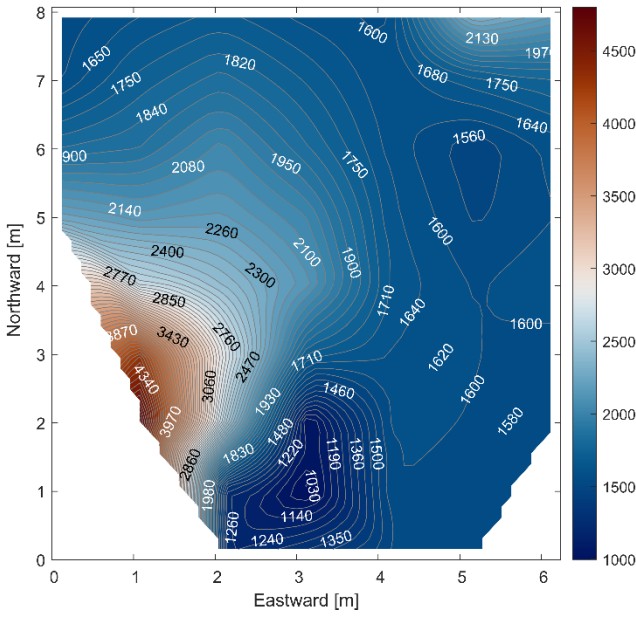

**Figure 4. Magnetic field intensity [nT] measured over the Mirador mine on 23 August 2019, using a 35,000 nT background. Elevated red values correspond to the steel balcony, while the large blue values indicate the mine entrance. Data points are spaced ~ 1 metre. Plots generated with MATLAB©. Credits: Yvelice Castillo.**

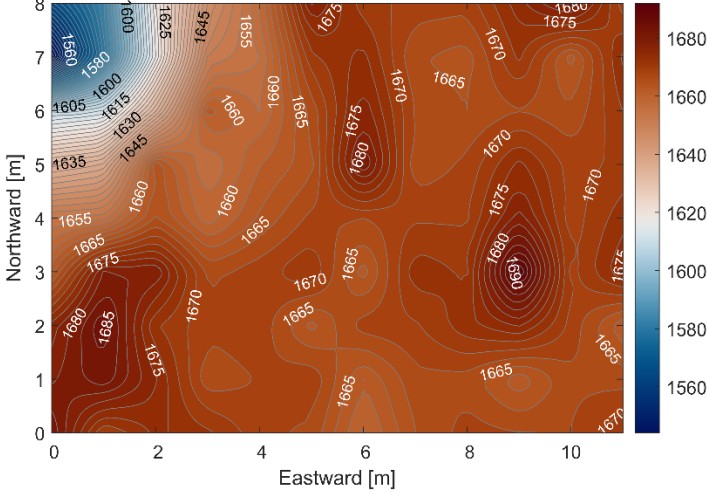

**Figure 5. Magnetic field intensity [nT] measured on the mountain above mine point number 6 on 13 November 2019, using a 35,000 nT background. The step size is about 1 metre. Plot generated with MATLAB©. Credits: Yvelice Castillo.**



### 3.4. The UNAH-Choluteca campus (13.326°, -87.135°) and the Centre for Aquaculture and Fisheries Research (CIAP, 13.4176494°, -87.4272342°, 6.25 masl), *Choluteca* province

On 6 May 2021, an aerial magnetic survey was conducted at CIAP, UNAH-Choluteca campus, in Choluteca province, with support from engineer Miguel Ángel García and his team at ENEE's Small Hydropower Unit. A Matrice 600 Pro drone was supplied, and Sulamith Kastl (BGR)provided a MagDrone SENSYS©. Table 3 summarises the core information from the survey.

**Table 3. CIAP's aerial prospecting core information.**

| # | Date | File name | Configuration (2 m steps) | Number of points |
|---|------|-----------|---------------------------|------------------|
| 1 | 2021/05/06 | 20210730_182144_MD-R3_#0042 | NWSE – SWNE mesh | 29522 |

The site was discarded due to:

● Large anomalies detected during prospecting.

● moderate pedestrian activity (anthropogenic noise).

### 3.5. The Army's First Communications Battalion, *Las Mesas, San Antonio de Oriente, Francisco Morazán* (14.0347°, -86.94°)

On 23 February 2022, a meeting at the Honduran Army's Joint Staff headquarters in Comayagüela, brought together Yvelice Castillo, Norman Palma, Manuel Rodríguez, Vilma Lorena Ochoa (then Dean of the UNAH's School of Space Sciences), Colonels Sauceda Sierra, Raúl López Coello, José Leandro Flores, and Sub-Lieutenant Carlos Martínez. Christopher Turbitt, Manager of Magnetic Observatories at the British Geological Survey (BGS), attended virtually. It was agreed to prepare a project profile for the first magnetic observatory in Honduras, to be presented to the Army's leadership.

On 10 March 2022, UNAH and Army representatives inspected both the Army's Defence School and the First Communications Battalion, ultimately selecting the latter, located in *Las Mesas* (38 km from Tegucigalpa on the road to *Danlí*). Plans included sourcing , designing the electrical supply and buildings, conducting topographic and magnetic surveys,  and training personnel. For the Army, the observatory was strategically important for maritime and aerial navigation, particularly offshore

Due to the COVID-19 quarantine, UNAH's magnetometer malfunctioned after batteries were left installed for over a year. PROGEO loaned a GEM GSM-19G Overhauser gradiometer, which was used on 27 April  2022 to survey an area of approximately 30 x 30 m (see Figure 12 in Annexes). The survey was conducted by Manuel Rodríguez, Felix Rodríguez, Norman Palma, and Yvelice Castillo.



A topographic survey was performed on 26 April 2022, by MSc. Marcela Norori (Department of Science and Technology of
Geographic Information, FACES/UNAH) and her students. They employed a SOUTH NTS-362 total station with angular
accuracy of ±2" and a distance-meter (2 mm + 2 ppm x D) with one prism.

On 26 June 2022, a larger polygon was surveyed using the BGR's MagDrone SENSYS© and ENEE's Matrice 600 Pro drone
(see Figure 13). Sites with lower anomalies were selected for follow-up ground surveys, which were delayed until 2024 due
to instrument breakdowns and lack of funds.

In August 2024, the team obtained a research grant from UNAH's Directorate of Scientific, Humanistic, and Technological
Research (DICIHT). Magnetic prospecting and site assessments were carried out between 8 August and 13 September 2024
(Table 4; surveys Figures 6, 7(a), and 7(b)). Although some promising zones were found, anomalies persisted, were, attributed
to terrain relief, anomalous bodies, and ferrimagnetic materials (hematite, ferrite).

**Table 4. The First Communications Battalion prospecting core information**

| # | Date | File name | Configuration (2 m steps) | Number of points |
|---|------|-----------|---------------------------|------------------|
| 1 | 2022/04/27 | TOTAL | NS – EW mesh | 720 |
| 2 | 2022/06/22 | 2.20220622_152909_MD-R3_#0042 (AERIAL) | NS – EW mesh | 528472 |
| 3 | 2022/06/22 | 1.20220622_145623_MD-R3_#0042 (AERIAL) | NS – EW mesh | 507832 |
| 4 | 2024/08/08 | SITIO | NS – EW mesh | 605 |
| 5 | 2024/08/23 | 3 | N – S profile | 68 |
| 6 | 2024/08/23 | 5 | WWS – EEN mesh | 362 |
| 7 | 2024/09/10 | 6 | WWS – EEN mesh | 369 |
| 8 | 2024/09/11 | 7 | WWS – EEN mesh | 346 |
| 9 | 2024/09/12 | 10 | NNW – SSE profile | 303 |
| 10 | 2024/09/13 | 11 | N – S profile | 108 |
| 11 | 2024/09/13 | 12 | N – S profile | 94 |
| 12 | 2024/09/13 | 13 | E – W profile | 85 |





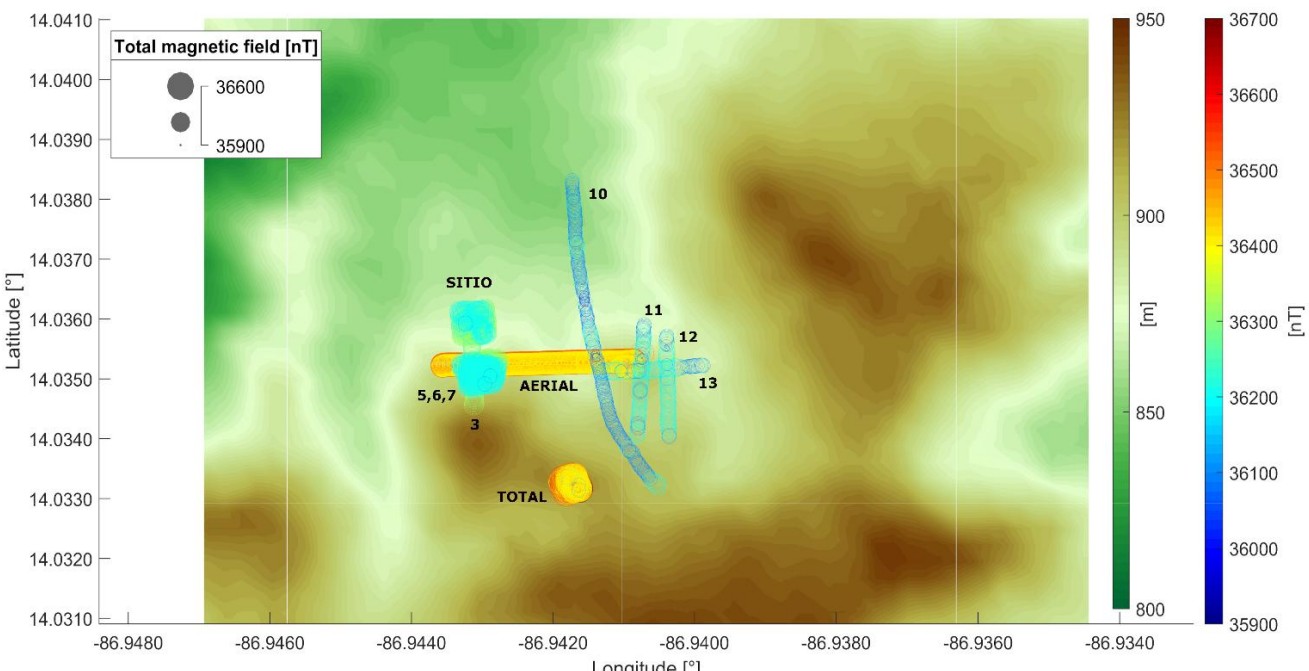

**Figure 6. Total magnetic field intensity [nT] measured in the First Communications Battalion in 2022 and 2024. Background raster image created with Open Topography data (**SRTM-NASA**, 2013). Map generated with Matlab©. Credits: Yvelice Castillo.**





(a)  (b)

**Figure 7. (a) Magnetic field intensity [nT] interpolation (colour bar) on a 35,000 nT background with elevation contours (grey dashed lines with magenta labels), computed from the *Las Mesas* 2024 ground prospecting SITIO, 3, 5, 7. (b) Similar interpolation for ground prospecting profiles 10, 11, 12, 13. Plots generated with Matlab©. Credits: Yvelice Castillo.**

Iván Guerrero notes in his geological report the *Morocelí* 2858 IIIG cartographic sheet identifies the area of interest as the
*Padre Miguel* formation, which consists of volcanic rocks, primarily pyroclasts. During the field investigation, tuff and lahar



deposits were distinguished, corroborating the data from the sheet. Additionally, there are indicated faults that may contribute
to the magnetic anomalies observed in the readings. Below the *Padre Miguel* formation, the red layers of the *Valle de Angeles*
group were identified, leading to an estimated maximum thickness of *Padre Miguel* of 100 m (data verified during the
geological inspection).

The *Las Mesas* battalion was discarded for the following reasons:

● persistent anomalies in all prospecting
● presence of ferrimagnetic materials

**3.6. The *Francisco Morazán* Power Station, *Cortés* province (15.035°, -87.756)**

Between 26 and 27 September, several sites were inspected inside tunnels and along both banks of the *Humuya* River, near
ENEE facilities. Four profiles were surveyed (Table 5). Figure 8 shows the P2 collection centre pathway, while Figure 9
presents an interpolation of all data. The limestone mountain displayed acceptable gradients (~ 1 nT/m), but the collection
centre showed large anomalies, probably due to buried materials.

**Table 5. The *Francisco Morazán* Power Station prospecting core information**

| # | Dates | Site name | File name | Prospecting (2 m steps) | Number of points |
|---|-------|-----------|-----------|------------------------|------------------|
| 1 | 2024/09/26 | The mountain at the east of the power station | 14a | E – W profile | 25 |
| 2 | 2024/09/27 | Pathway to the collection centre (P2) | 14b | E – W profile | 45 |
| 3 | 2024/09/27 | Collection centre (P2) | 15 | SSW – NNE profile | 56 |
| 4 | 2024/09/27 | Mountain north of P2 | 16 | EES – WWN profile | 26 |

The site was discarded because:

● vandalism  was reported at the P2 collection centre;

● 50 Hz signals from the power station transformers may interfere with observatory measurements.





**Figure 8. Total magnetic field intensity [nT] measured in the P2 collection centre at *El Cajón* in 2024. Background raster image created with Open Topography data (SRTM-NASA, 2013). Map generated with Matlab©. Credits: Yvelice Castillo.**





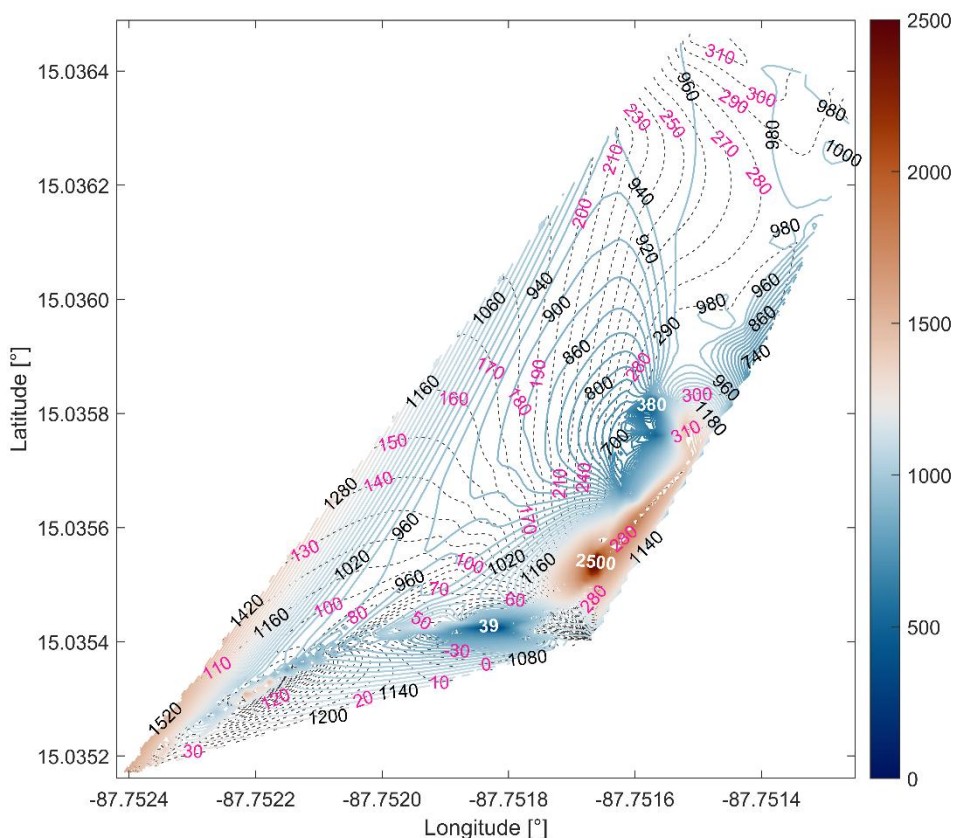

**Figure 9. Magnetic field intensity interpolation [nT] on a 36,000 nT background (colour bar) with elevation contours [m.a.s.l.] (dashed lines and magenta labels), computed from 2024 data collected at the *El Cajón* collection centre. Larger and lower magnetic values may correspond to buried materials. Plots generated with Matlab©. Credits: Yvelice Castillo.**

### 3.7. The Second Infantry Battalion in *Támara, Francisco Morazán* (14.005º, -87.016º) and the First Engineers Battalion in *Siguatepeque, Comayagua* province (14.188°, -87.333°)

The most isolated parts of both battalions were inspected, but they failed to meet the required isolation conditions (~380 m

free of infrastructure and traffic). Consequently, no surveys were conducted. In *Támara, the* Army's *Tesón* survival program

created continuous human activity, while in *Siguatepeque* to battalion is too close to the urban centre.



### 3.8. The First Artillery Battalion in *Zambrano, Francisco Morazán* province (14.006º, -87.006º)

The *Zambrano* Mountain, located northeast of the First Artillery Battalion, was selected to install the magnetic observatory. Table 6 summarises the surveys. The Figure 10 shows the distribution of magnetic intensity, while Figure 11(a) presents interpolations of the 2024 -- 2025 data and the Figure 11(b) presents the southern section of this survey, i.e., the site selected to install the magnetic observatory. Larger values correlate with higher elevations.

**Table 6. *Zambrano* prospecting core information**

| # | Date | Site name | File | Prospecting (2 m steps) | Number of points |
|---|------|-----------|------|-------------------------|------------------|
| 1 | 2024/10/11 | South slope of the *Zambrano* Mountain | 19 | SE – NW profile | 31 |
| 2 | 2024/10/11 | South slope of the *Zambrano* Mountain | 20 | SE – NW profile | 29 |
| 3 | 2024/10/24 | South slope of the *Zambrano* Mountain | 21 | N – S mesh | 348 |
| 4 | 2025/03/14 | Top of the *Zambrano* Mountain | 10a | S – N profiles | 24 |
| 5 | 2025/03/14 | Top of the *Zambrano* Mountain | 10b | S – N profiles | 25 |
| 5 | 2025/04/04 | South slope of the *Zambrano* Mountain | 11a | N – S profile | 26 |
| 6 | 2025/04/04 | South slope of the *Zambrano* Mountain | 11b | N – S profile | 26 |
| 7 | 2025/04/04 | South slope of the *Zambrano* Mountain | 11c | N – S profile | 26 |
| 8 | 2025/04/04 | South slope of the *Zambrano* Mountain | 11d | N – S profile | 29 |
| 9 | 2025/04/11 | South slope of the *Zambrano* Mountain | 12a | N – S profile | 127 |
| 10 | 2025/04/11 | South slope of the *Zambrano* Mountain | 12b | N – S profile | 115 |
| 12 | 2025/04/25 | South slope of the *Zambrano* Mountain | 13 | N – S mesh | 115 |



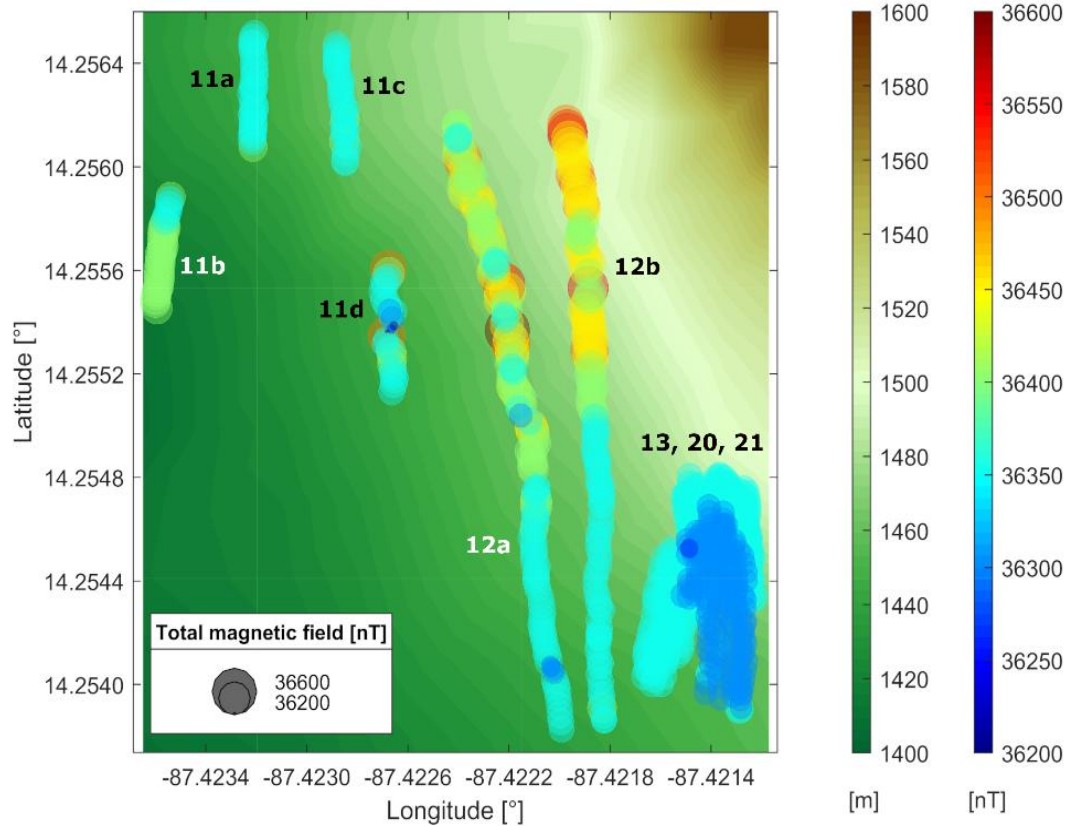

**Figure 10. Total magnetic field intensity [nT] and files distribution in *Zambrano* Mountain. Background raster image created with Open Topography data. Map drawn with Matlab©. Credits: Yvelice Castillo.**





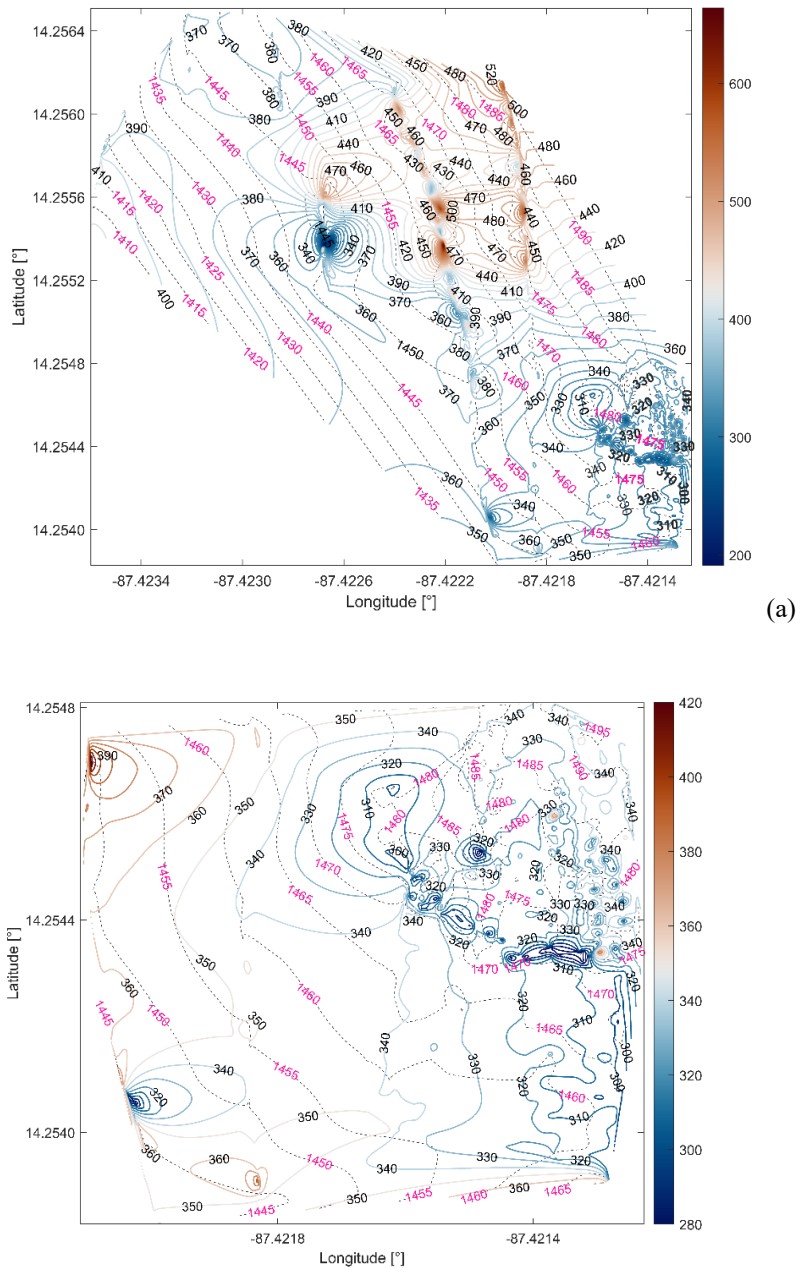

**Figure 11. (a) Magnetic field intensity interpolation [nT] on a 36,000 nT background (colour bar) and elevation contours [m.a.s.l.] (dashed lines with magenta labels), generated from the *Zambrano* prospecting data: 11a, 11b, 11c, 11d, 12a, 12b, 13, 20 and 21. (b) Detail of the south-eastern section (prospectings 13, 20, 21 and part of 12a and 12b), showing the selected area for the magnetic observatory. Plots generated with Matlab©. Credits: Yvelice Castillo.**




Key results of the Zambrano prospecting include:

- adequate security and no pedestrian traffic during surveys;

- anomalies mainly associated with rocks;

- a primary power line crossing to the north of the area;

- steep terrain and difficulty in transporting materials;

- The presence of a small brick hut near profile 12a, suitable for installing a variometer or scalar magnetometer.

### 4.4. UNAH's projects linked to MAGHO

In 2023, M.Sc. Miguel Angel Rojas (Technological Institute of Costa Rica) coordinated with the Upper Atmosphere Laboratory of the University of Texas at Dallas to install two ScintPi© systems in Honduras. These systems monitor the 1.2 GHz and 1.6 GHz signals from GNSS constellations to map local ionospheric scintillation and total electron content over Tegucigalpa.

In February 2025, thanks to Dr. Pedro Corona, a researcher at the Space Weather Service of Mexico (SCiESMEX), the Johns Hopkins University Applied Physics Laboratory donated an EZIE-Mag© triaxial magnetometer for installation in Tegucigalpa. This equipment is intended to be installed in the MAGHO's site, aiming to establish a space weather observatory.

The acquisition of a Cherenkov detector and two *Cosmic Watch* muon detectors are being negotiated through the Latin American consortium El Bongó and the Latin American Giant Observatory (LAGO) project.

### 4.5 Summary of results

- *La Tigra* mines and mountains presented significant anomalies due to ore bodies, as well as high humidity and security concerns.

- CIAP and *La Tigra* do not meet the required magnetic cleanliness requirements.

- Las Mesas showed persistent anomalies likely caused by terrain relief and ferromagnetic materials.

- The limestone terrain at the *Francisco Morazán* Power Station showed promising gradients, but vandalism and 50 Hz electromagnetic interference made the site unsuitable





●   *Zambrano* Mountain showed acceptable results, though anomalies in rocks and logistical challenges remain. Some
concerns exist regarding the difficulty of getting materials up, whether rocks with large anomalies can be removed,
and maybe some interference of the primary line crossing at the north of the surveys. A small brick hut there could
be used to install one magnetometer.

## 5. Conclusions

●   The "Tigra" mines and surrounding mountains revealed significant anomalies caused by ore bodies, in addition to
high humidity and low security, making them unsuitable.

●   Neither CIAP nor La Tigra met the required magnetic cleanliness conditions.

●   Prospecting at the First Communications Battalion in Las Mesas revealed persistent anomalies, likely associated with
terrain relief, slopes, anomalous bodies, and the presence of hematite, magnetite, or other ferromagnetic materials.

●   The limestone soils around the Francisco Morazán Power Station produced the most promising results, with gradients
of ~1 nT/m achievable if buried materials were removed. However, reported vandalism and probable 50 Hz
electromagnetic interference from the power station transformers make this site unreliable for observatory operations.

●   The Zambrano Mountain, particularly its southern slope, presented acceptable conditions. The correlation between
magnetic gradients and elevation was evident. Nonetheless, some challenges remain regarding the difficulty of
transporting construction materials, the presence of rocks with large anomalies, and possible interference from a
nearby primary power line. The small brick hut located near the southern end of profile 12a could serve as an initial
installation point for one variometer.

## 6. Recommendations

The experience gained throughout this work is invaluable for implementing IAGA and INTERMAGNET standards for
magnetic observatory installations. Based on our findings, we recommend the following:

1.   Infrastructure and collaboration

The construction of at least three huts and two non-magnetic pillars is necessary for installing the magnetometers. Support
should be sought from the Honduran Army, the National Electrical Energy Company (ENEE), and international agencies.
Opportunities for donations of instruments and materials should be actively pursued.

2.   Site development

Further assessment of Zambrano Mountain is recommended to determine whether anomalies caused by rocks can be
mitigated and whether the challenges of material transport and power line interference can be addressed.

3.   Integration with space weather research



The establishment of a space weather observatory managed by UNAH is strongly advised. This facility should integrate:

■  the planned magnetic observatory,

■  the GNSS receiver network for ionospheric studies,

■  EZIE-Mag network,

■  Latin American Giant Observatory facilities, and

■  other associated projects.

Such an integrated observatory would not only contribute high-quality geomagnetic data but also strengthen regional capacity

for monitoring and forecasting space weather hazards.




# 7. Annexes

## 7.4. First Communications Battalion magnetic prospecting

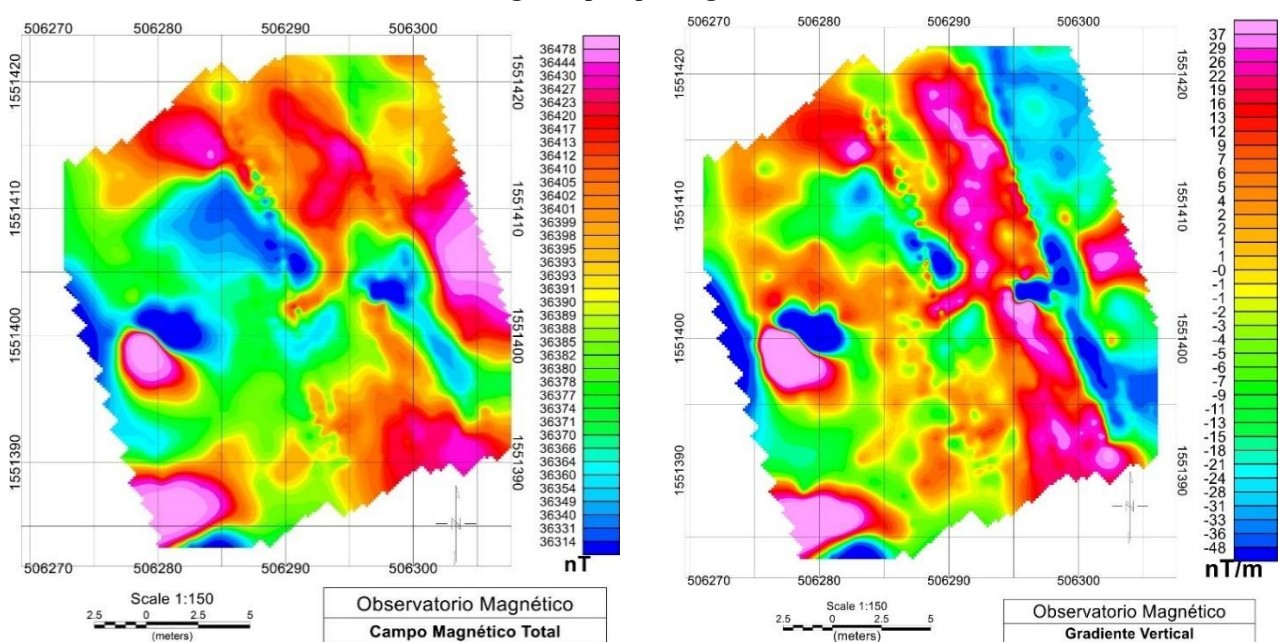

**Figure 12.** *Las Mesas'* mesh (file *TOTAL*) on April 27, 2022. Left: total magnetic field intensity [nT]. Right: vertical magnetic field intensity gradient [nT/m]. Plots generated with Oasis Montaj© software. Credits: PROGEO and Manuel Rodríguez.

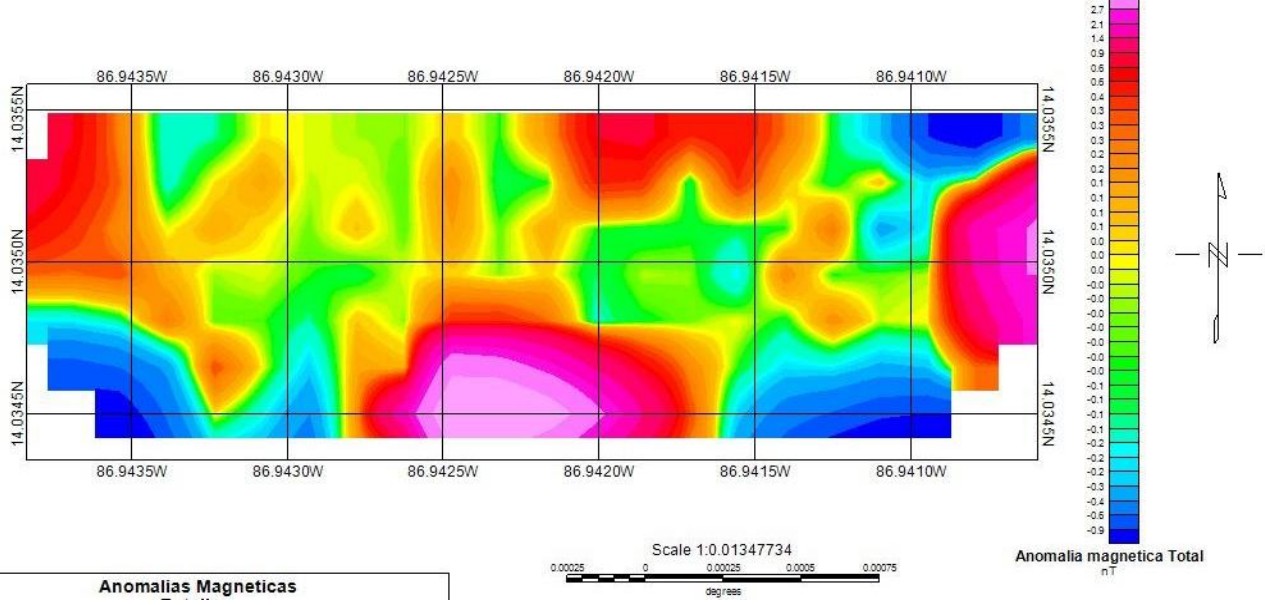

**Figure 13.** *Las Mesas'* aerial magnetic anomaly obtained with the BGZ's Magdrone SENSYS© on June 26, 2022. Drone elevation and speed: ~29 m and 4.32 km/h. Plot generated with Oasis Montaj© software. Credits: PROGEO and Manuel Rodríguez.





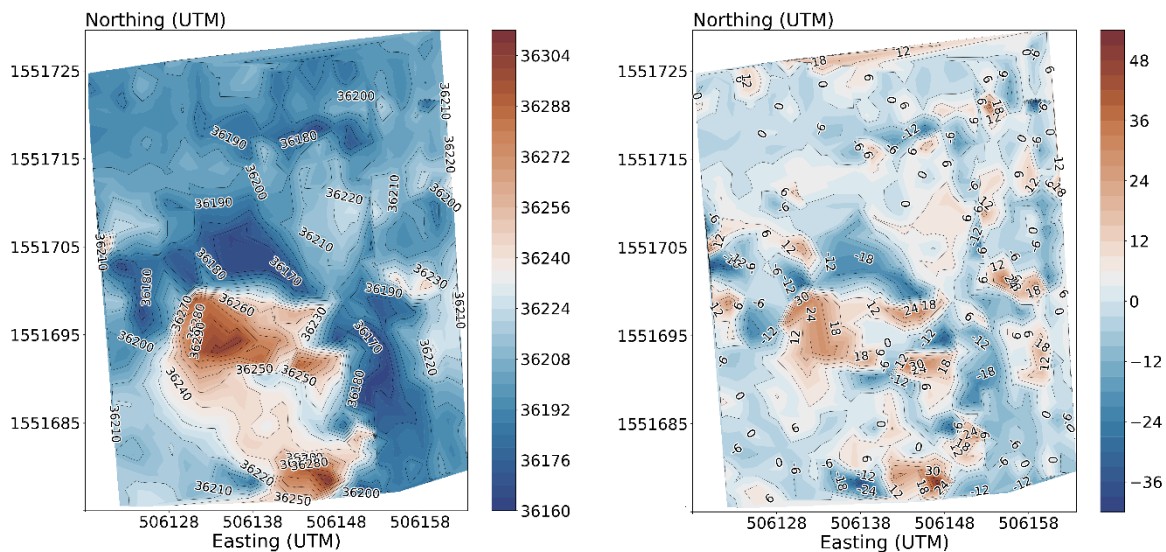

**Figure 14.** *Las Mesas'* **mesh (file 'SITIO') on August 8, 2024. Left: total magnetic field intensity [nT]. Right: vertical magnetic field gradient [nT/m]. ~2m steps. Plots generated with SciServer©, a resource of the Institute for Data Intensive Engineering and Science at Johns Hopkins University (IDIES). Credits: André Aguilar, Carlos Osorio, Isaías Martínez, Jonathan Vides, Oscar Mendieta, Samuel Flores, Yvelice Castillo.**

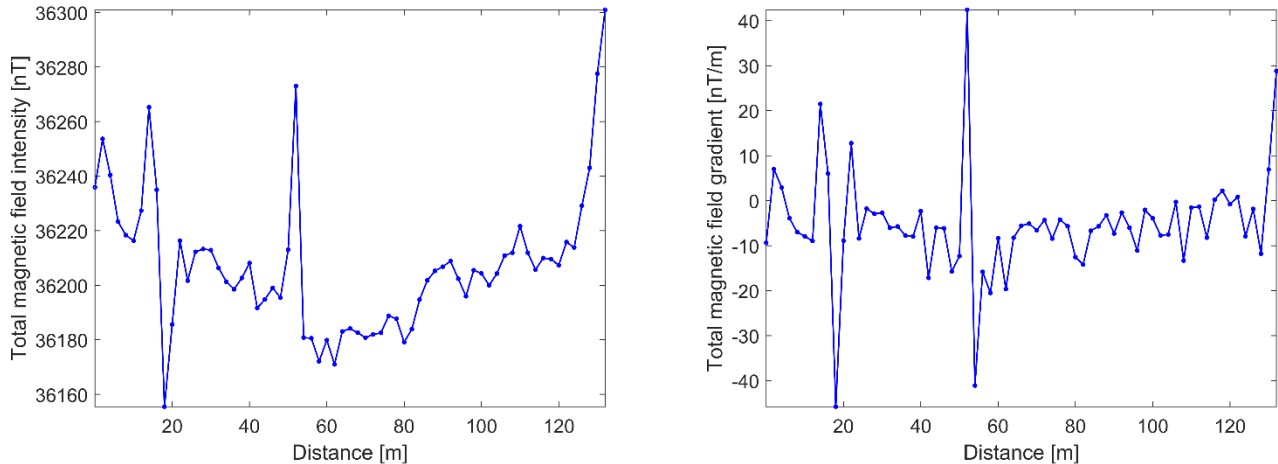

**Figure 15.** *Las Mesas'* **N – S profile #3 on August 23, 2024. Left: total magnetic field intensity [nT]. Right: vertical magnetic field gradient [nT/m]. ~2 m steps. Plots generated with Matlab©. Credits: Yvelice Castillo.**



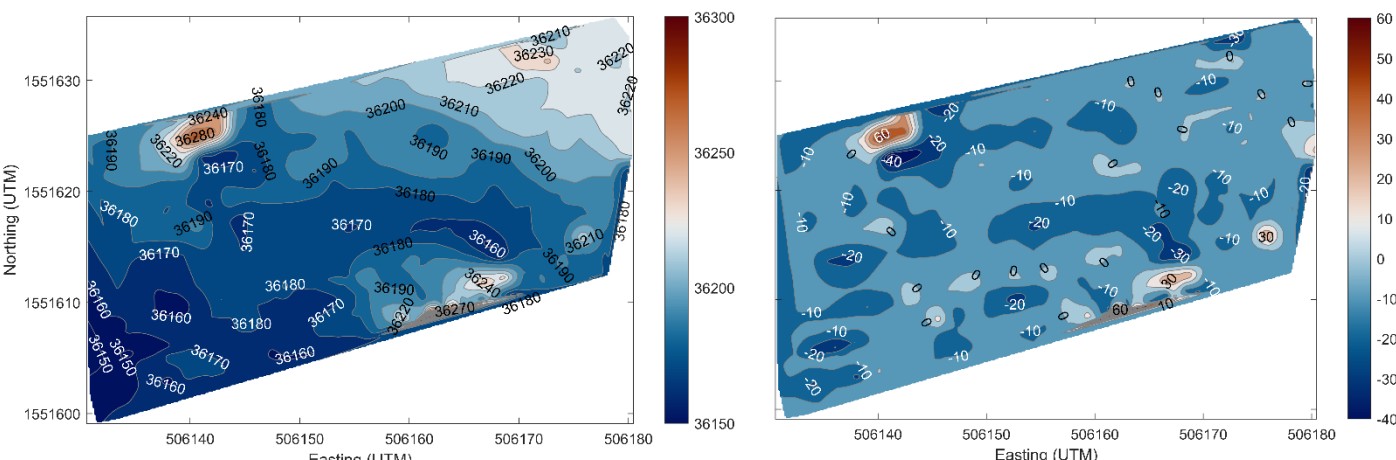

**Figure 16.** *Las Mesas'* **mesh #5 on August 23, 2024. Left: total magnetic field intensity [nT]. Right: vertical magnetic field gradient [nT/m]. ~2m steps. Same credits and resources as in the previous figure.**

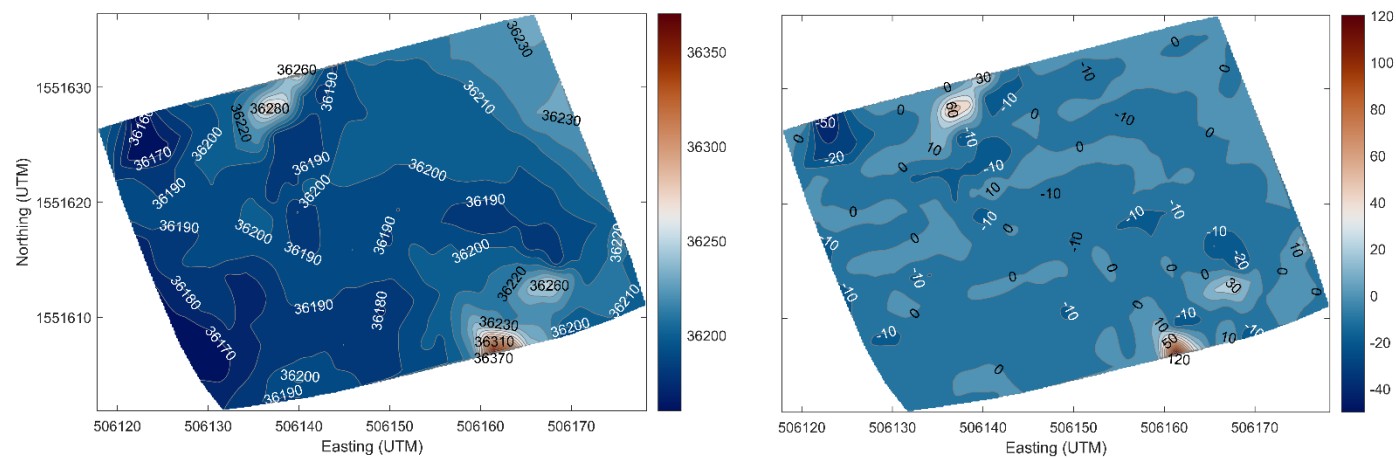

**Figure 17.** *Las Mesas'* **mesh #6 of September 10, 2024. Left: total magnetic field intensity [nT]. Right: vertical magnetic field gradient [nT/m]. ~2m steps. Same credits and resources as in the previous figure.**





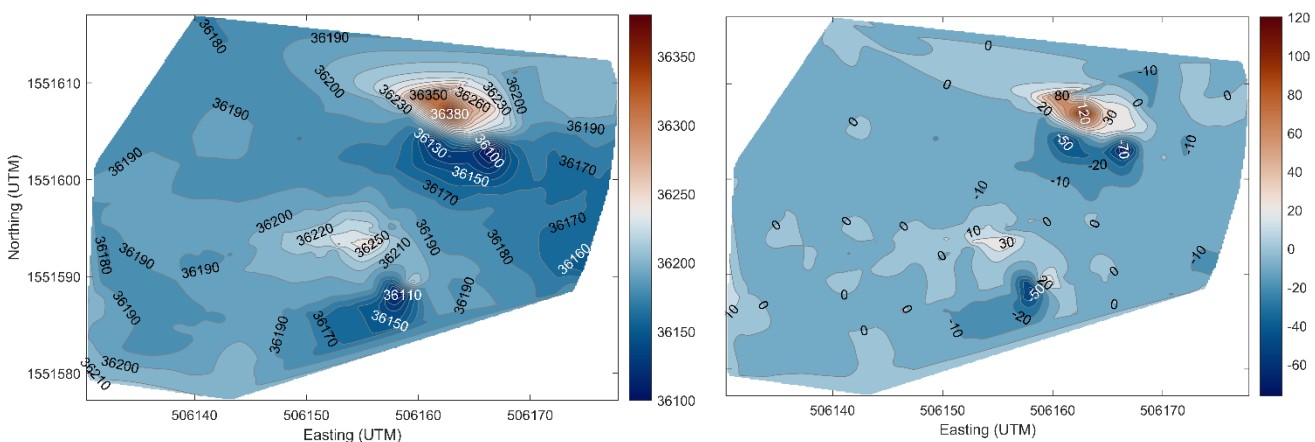

**Figure 18.** *Las Mesas'* **mesh #7 on September 11, 2024. Left: total magnetic field intensity [nT]. Right: vertical magnetic field gradient [nT/m]. ~2m steps. Same credits and resources as in the previous figure.**

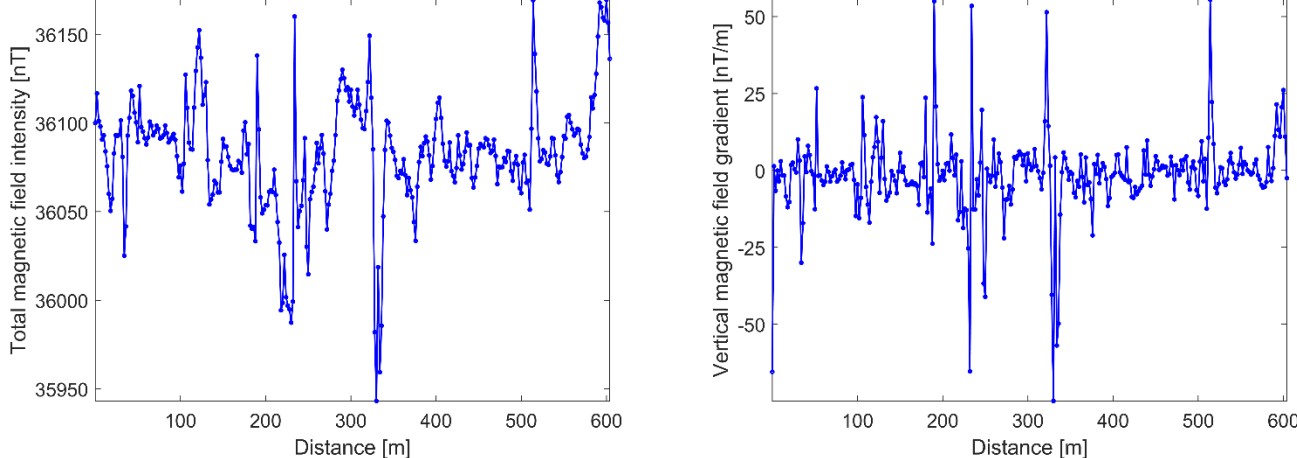

**Figure 19.** *Las Mesas'* **NNW—SSE profile #10 on September 12, 2024. Top: total magnetic field intensity [nT]. Bottom: vertical magnetic field gradient [nT/m]. ~2m steps. Same credits and resources as in the previous figure.**



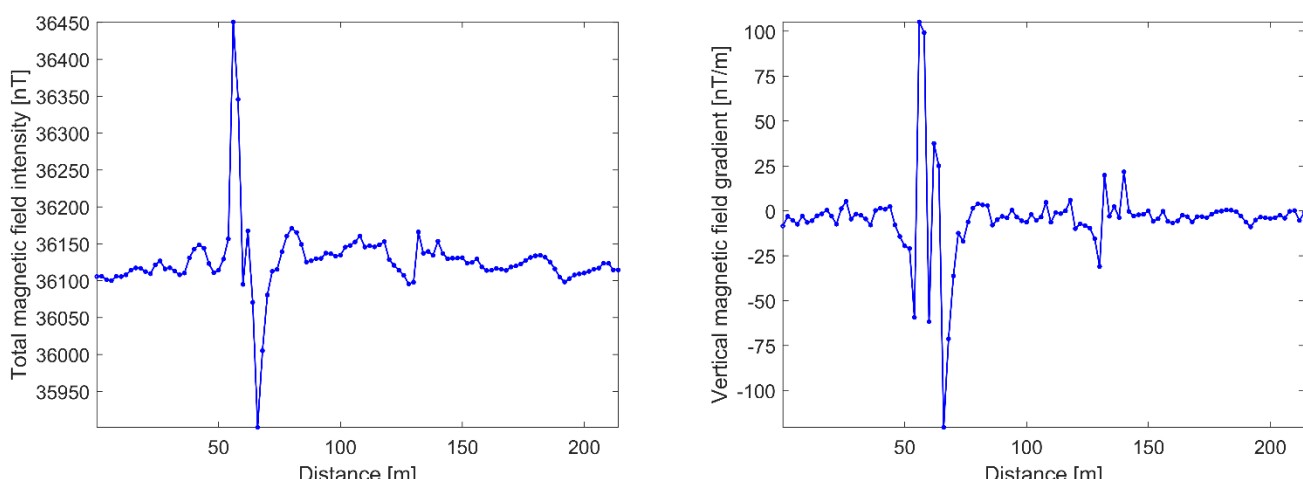

**Figure 20.** *Las Mesas'* N – S profile #11 on September 13, 2024. Left: total magnetic field intensity [nT]. Right: vertical magnetic field gradient [nT/m]. ~2m steps. Same credits and resources as in the previous figure.

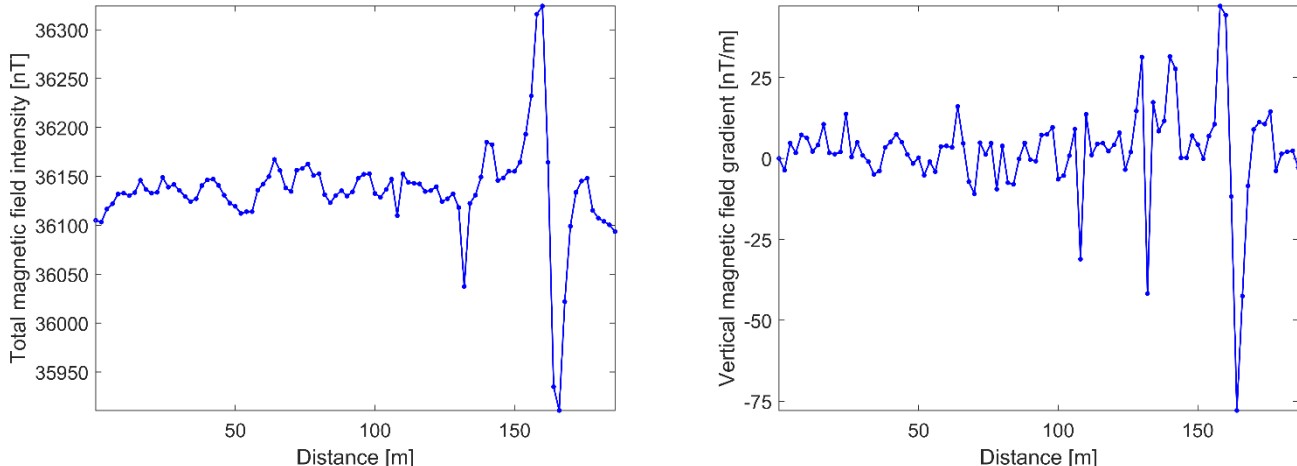

**Figure 21.** *Las Mesas'* N – S profile #12 on September 13, 2024. Left: total magnetic field intensity [nT]. Right: vertical magnetic field gradient [nT/m]. ~2m steps. Same credits and resources as in the previous figure.




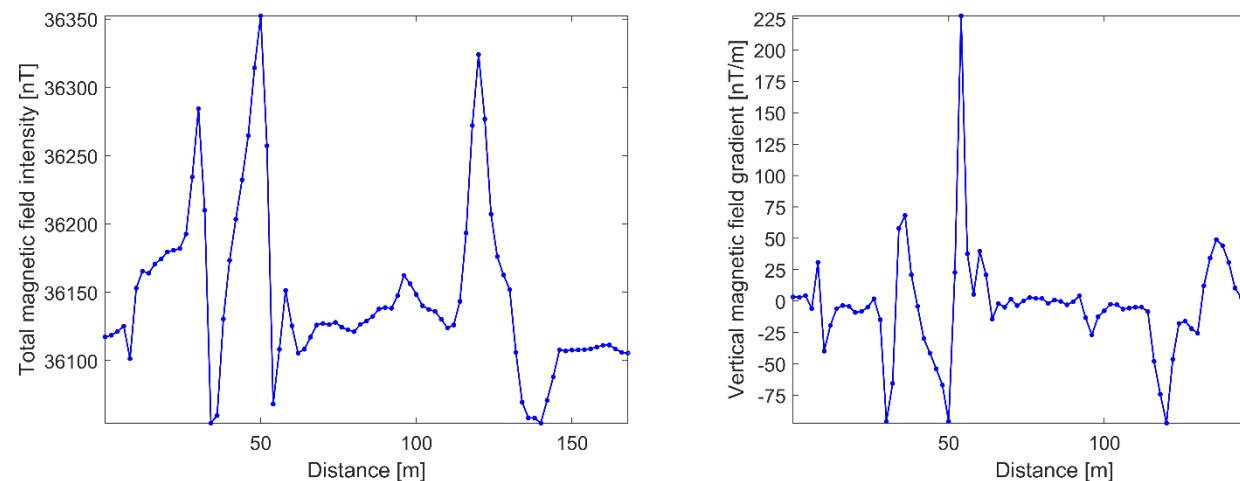

**Figure 22.** *Las Mesas'* E – W profile #13 on September 13, 2024. Left: total magnetic field intensity [nT]. Right: vertical magnetic field gradient [nT/m]. ~2m steps. Same credits and resources as in the previous figure.

**7.5.  Prospecting in the *Francisco Morazán* Power Station**

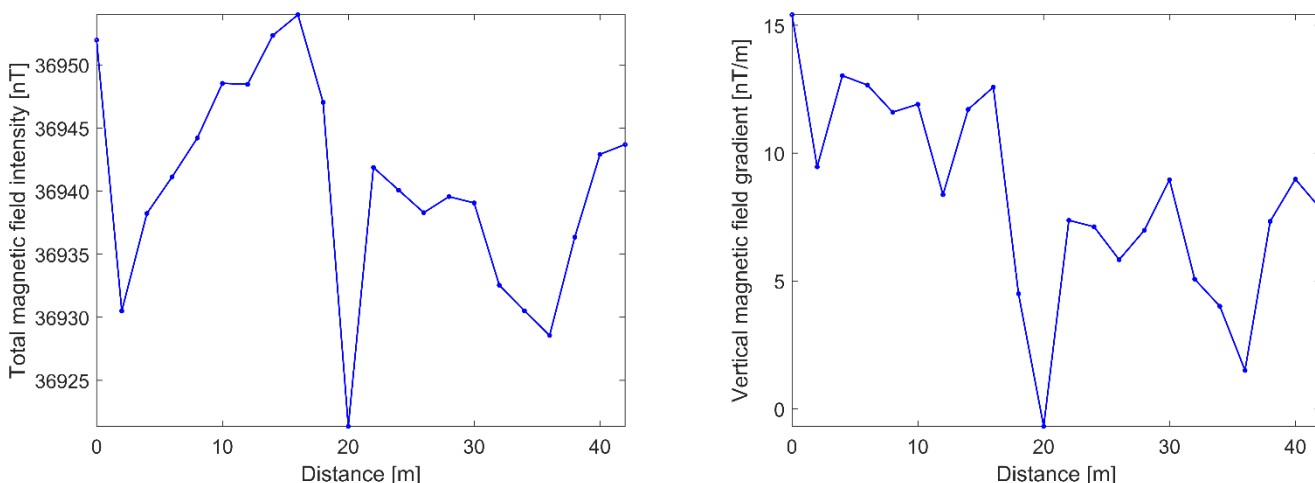

**Figure 23. Profile #14a in a mountain at the NE of the power station. Left: total magnetic field intensity [nT]. Right: vertical magnetic field intensity gradient [nT/m]. ~2m steps. Measured on September 26, 2024. Same credits and resources as in the previous figure.**





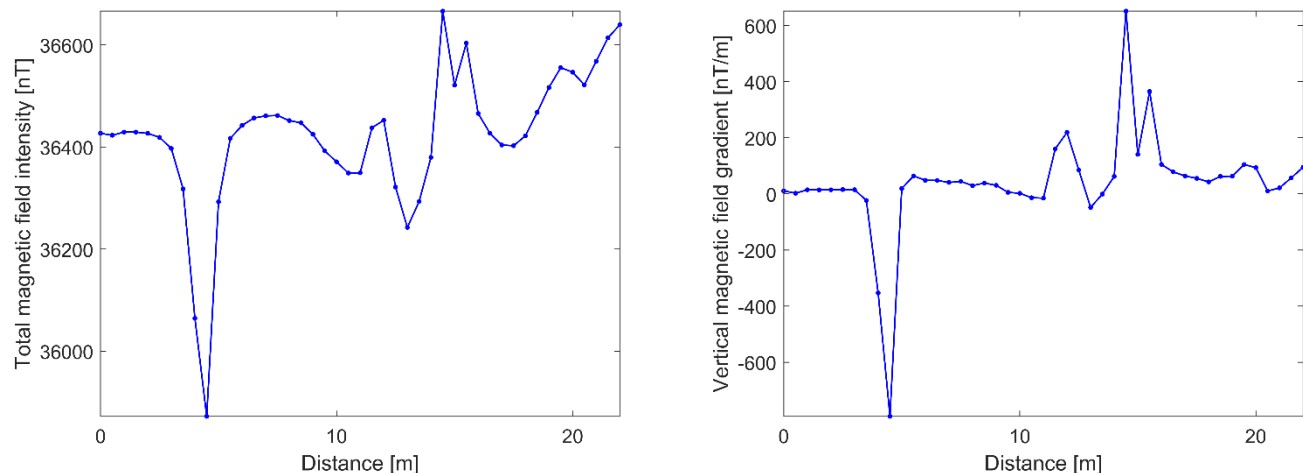

**Figure 24. Profile #14b along the pathway east of P2 on September 27, 2024. Left: total magnetic field intensity [nT]. Right: vertical magnetic field intensity gradient [nT/m]. ~2m steps. Large anomalies must be due to buried ferromagnetic materials. Same credits and resources as in the previous figure.**

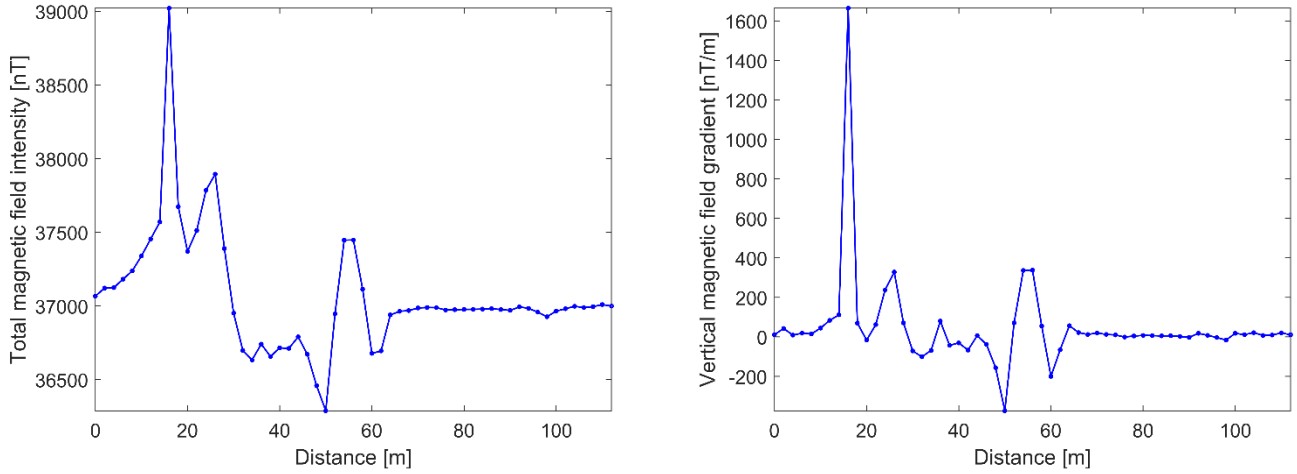

**Figure 25. Profile #15 from the pathway to the mountain at the north of P2 on September 27, 2024. Left: Total magnetic field intensity [nT]. Right: Vertical magnetic field gradient [nT/m]. ~2m steps. Significant anomalies must be due to buried ferromagnetic materials—same credits and resources as in the previous figure.**

445



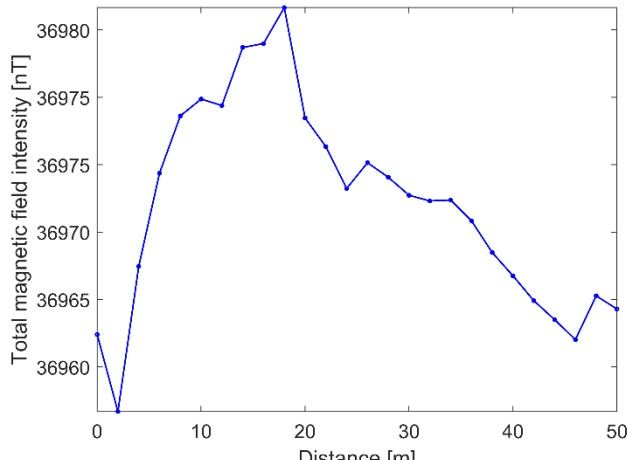
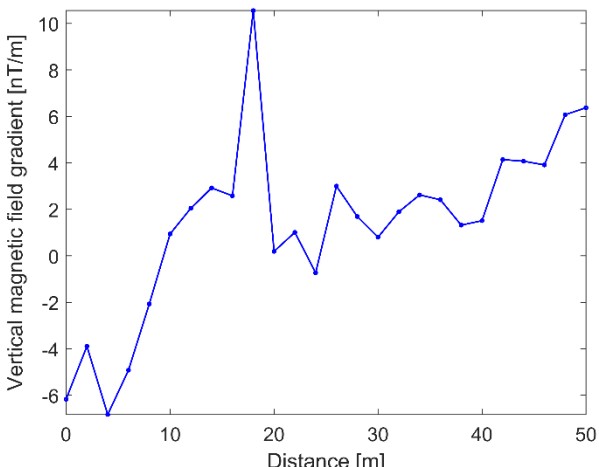

**Figure 26. Profile #16 in the mountain north to P2 on September 27, 2024. Left: Total magnetic field intensity [nT]. Right: Vertical magnetic field gradient [nT/m]. ~2m steps. Same credits and resources as in the previous figure.**

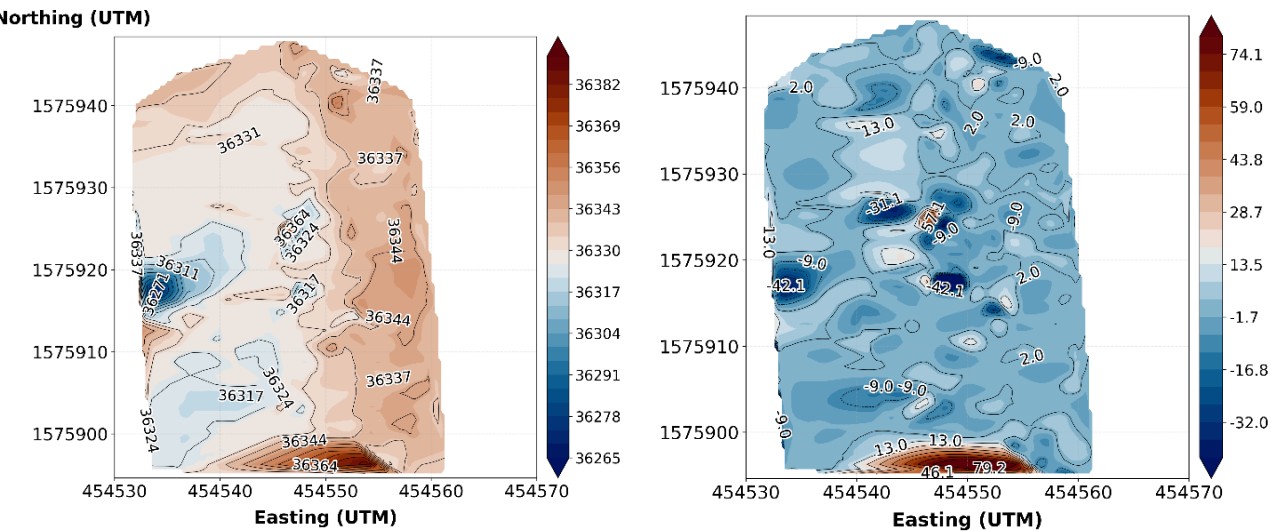

**Figure 27. Mesh #21 at the SE of the *Zambrano* Mountain on October 24, 2024. Left: Total magnetic field intensity [nT]. Right: Vertical magnetic field gradient [nT/m]. ~2m steps. Plots generated with SciServer©, a resource of the Institute for Data Intensive Engineering and Science at Johns Hopkins University (IDIES). Credits: André Aguilar, Carlos Osorio, Isaías Martínez, Jonathan Vides, Oscar Mendieta, Samuel Flores, Yvelice Castillo.**

446





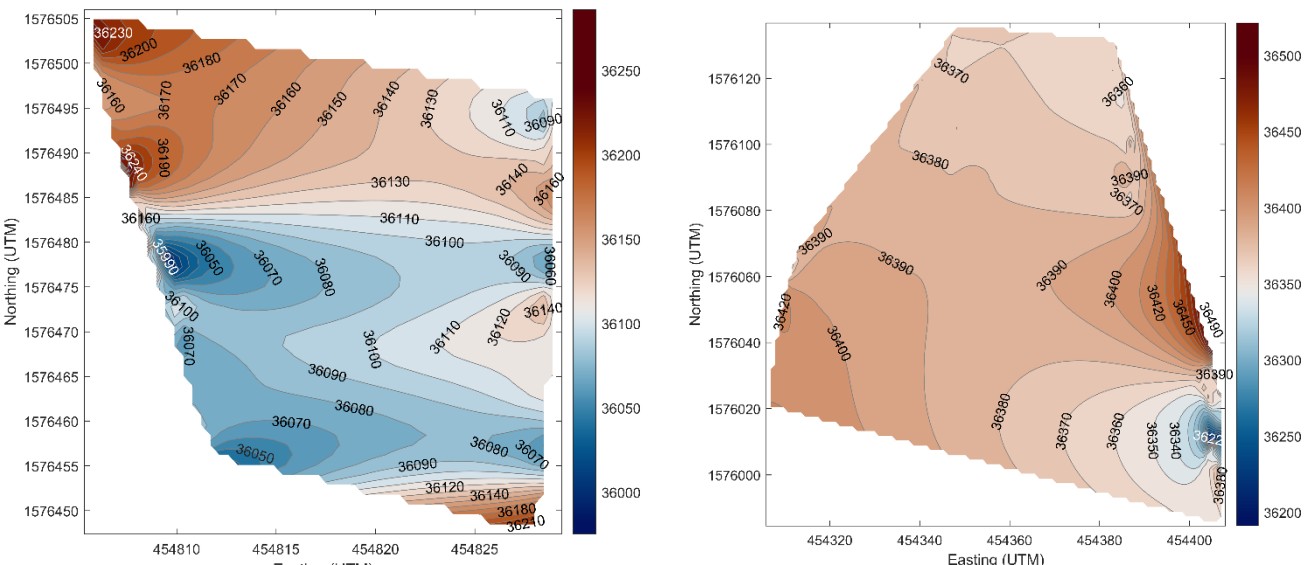

**Figure 28. Left: Total magnetic field intensity [nT] interpolation at the top of the *Zambrano* Mountain, measured on March 14, 2025 (profiles 10a and 10b). Right: Total magnetic field intensity [nT] interpolation on April 4, 2025 (profiles 11a, 11b, 11c, 11d). Plots drawn by Yvelice Castillo using Matlab©.**

447





**Figure 29. Left: Total magnetic field intensity interpolation [nT] computed with the April 11, 2025's data (profiles 12a and 12b) of the *Zambrano* Mountain. Right: similar interpolation done with the April 25, 2025, data (profile #13). Irregular steps (~2m). Plots drawn by Yvelice Castillo using Matlab©.**





448

**8. Data availability**

Most of the original data is available in the Research Gate network.

451

**9. Supplement link: https://www.researchgate.net/lab/Observatorio-Magnetico-de-Honduras-MAGHO-Yvelice-Soraya-Castillo-Rosales**

454

10. **Author contribution**

11. Y. C. acted as project administrator and prepared the manuscript with contributions from all co-authors. She also drafted the funding proposal and technical reports. Y. C., N. P., M. R., F. R., and I. G. carried out the ground prospecting. C. T., J. Ri., J. Ra., and N. G. provided specialised advice. G. C. and A. C. contributed with expert training. Y. C. produced most of the figures. The remaining authors contributed to field prospecting, data acquisition, and figure preparation.

12. **Competing interests**

The authors declare that they have no competing interests.

13. **Acknowledgements**

We gratefully acknowledge UNAH's Rector, Ph.D. Odir Aarón Fernández, and Ph.D. Ricardo Matamoros, Director of UNAH's Scientific, Humanistic and Technological Research Directorate (DICIHT), for granting the budget under substructure 2-05-09-02 *Fondo Concursable para la Investigación Científica y Tecnológica*, which enabled the 2024 activities.
We also thank:
- Javier Mejuto, Dean of FACES/UNAH;
- Vilma Ochoa, former Dean of the Faculty of Space Sciences (FACES/UNAH);
- Nohemy Rivera, Director of the Institute of Archaeoastronomy, Cultural and Natural Heritage (UNAH);
- Martha Talavera, Head of the Department of Astronomy and Astrophysics;
- Lidia Torres, former Director of the Honduran Institute of Earth Sciences (UNAH Faculty of Sciences);
- General Roosevelt Hernández, Army Joint Staff Chief;
- Eduardo Gross, former Dean of the Engineering College;
- Marta Castro, former Chief of UNAH's Civil Engineering Laboratories;
- Carlos Luis Barahona, Technician of the Department of Astronomy and Astrophysics;
- Hugo Heomar Ramos Hernández, Teacher at the Department of Astronomy and Astrophysics;.



- General Walter Amador Lacayo; Colonels Wilfredo Oseguera, Roger Oseguera, Sauceda Sierra, Raúl López Coello, José Leandro Flores, Salguero, and Denis Omar Velásquez;

- Mayor Fredy Valdez Niño; and Second Lieutenant Carlos Martínez.

Special recognition is due to engineer Maryuri García, Johana Marcela Norori and her students from the CTE-123 Topography and Laboratory course, Luis Fonseca, Jorge Morazán, Miguel García, and their collaborators for their invaluable support. We are grateful to the ENEE for lending the Matrice 600 Pro drone, and to Sulamith Kastl (BGR) for providing the MagDrone SENSYS©.

We further acknowledge Ing. Marliu Samael Alvarado, our guide at the Francisco Morazán Power Station, and Thomas Martyn (BGS) for their valuable advice.

Our gratitude extends to Maria Alexandra Pais, who conceived this project, as well as João Fernandes and colleagues at the Centre for Earth and Space Research of the University of Coimbra (CITEUC). Special thanks go to Paulo Ribeiro for training in data processing at the Magnetic Observatory of the University of Coimbra.

We also appreciate the contributions of Esteban Hernández (RIP), and Armando García from the Institute of Geophysics of UNAM, and the Pan-American Institute of Geography and History (UNAM), which awarded us a scholarship to participate in the Pan-American Workshop on Geomagnetism in Mexico (2018).

We acknowledge Iván Monge, Armando Ayala, and Jorge Brenes from the Costa Rican Institute of Electricity (ICE) and the Santa Elena Observatory for their guidance, as well as Luiz Benyosef and colleagues at the National Observatory of Brazil for supporting our participation in the II Pan-American Workshop on Geomagnetism (2017).

Special thanks to José Cáceres for his advice on using the Open Topography database and to Enrique Guerrero for reviewing the manuscript's grammar and style. We also thank the organisers of the XX IAGA Workshop and Summer School in Vassouras, Brazil (2024); Miguel Ángel Rojas (Instituto Tecnológico de Costa Rica); Fabiano Rodrigues and Josemaría Gómez-Sócola (University of Texas at Dallas); and Luis Alberto Núñez de Villavicencio, P. I. of the Latin American Giant Observatory (LAGO).

This research made use of SciServer©, IDIES Data Centre, a resource developed and operated by the Johns Hopkins University Institute for Data Intensive Engineering and Science (IDIES).

Finally, we sincerely thank the reviewers of this article for their valuable comments, which helped us improve the manuscript.

## 14. Financial support

This work was partially funded through Substructure 2-05-09-02, the Competitive Funds for Scientific and Technological Research of the Directorate of Scientific, Humanistic, and Technological Research (DICIHT) of the UNAH.





510

511

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
