# Peer review of "The first magnetic observatory of Honduras: Assessment and magnetic prospecting in 2019 - 2025"

_EGUsphere, 2025_

## Referee Comment (RC1)

**Report:** 'The first magnetic observatory of Honduras: assessment and magnetic prospecting in 2019-2025'

**by** Yvelice-Soraya Castillo-Rosales, Norman-Iván Palma-Cruz, Manuel-de-Jesús Rodríguez-Maradiaga, Félix-Enrique Rodríguez-Garciá, Iván-Jorel Guerrero-Mejía, Christopher-William Turbitt, André-Jared Aguilar-Ochoa, Carlos-Alberto García-Osorio, Oscar-Rolando Mendieta-Brizuela, Isaías-Rafael Martínez-Hernández, Samuel-Elías Flores-Portillo, Jonathan-Luciano Vides-Zerón, Jean Rasson, John Riddick, Gerardo Cifuentes-Nava, Ana Caccavari-Garza, Natalia Gómez-Pérez

GLOBAL COMMENTS

This study reports on an important effort to identify a suitable site for the installation of the first magnetic observatory in Honduras. The relevance of such an infrastructure is clearly explained, as it enables both the monitoring of the secular variation of the Earth's main magnetic field and the observation of geomagnetic activity associated with interactions with energetic particles from the Sun, which may pose risks to various types of man-made infrastructure.

Due to a range of difficulties encountered throughout the project—such as the need to coordinate with multiple institutions, limited funding for instrumentation, lack of prior experience in this field, the COVID-19 pandemic, and the presence of different sources of perturbations at the tested sites—the project extended over a period of six years.

Although the relevance of this type of study is fully acknowledged, the submitted manuscript largely reads as a report of the work carried out and lacks a deeper and more rigorous analysis of the data acquired during the various surveys. Moreover, even as a report, it lacks clarity in several points, as detailed below.

The manuscript also does not lead to a clear final choice for the new magnetic observatory. The apparently best candidate site, Zambrano Mountain, still requires additional testing, particularly with respect to the presence of magnetized rocks, the influence of a nearby power line, and various logistical challenges. In addition, electrical soundings would be required to properly characterize the lateral variability and depth profile of the local ground conductivity.

MAIN ISSUES IN THE MANUSCRIPT:

1- A geological and lithological characterization of the surveyed region is missing, including an assessment of its tectonic setting. Such information is essential for understanding the distribution of undesirably strong crustal magnetization and for identifying the most suitable locations within limestone terrains.

2- We understand that a great effort has been made over six years through multiple surveys. However, the manuscript gives too much emphasis to reporting all the given steps and lacks a more rigorous analysis of data that could yield results of broader interest to other teams facing similar challenges. For instance, in the case of the First Communications Battalion site, a substantial amount of survey data is available (including aerial and ground magnetic profiles as well as mesh surveys), covering an area of approximately 400 × 600 m. Why has no attempt been made to integrate these datasets into a single magnetic anomaly map?

3- In some cases, a surveyed site is dismissed not on the basis of the magnetic survey data presented in the manuscript, but rather on the basis of other information that is not

sufficiently discussed. For example, the rejection of the Mirador mine site appears to rely on the reportedly high humidity levels inside the mine. Were these values compared with those observed at established observatories, such as Conrad Observatory or observatories located in Arctic regions? Similarly, concerns related to the 50 Hz signal from the Francisco Morazán power station are mentioned. Was a variometer operated in a continuous monitoring mode over a sufficiently long period to properly assess this disturbance?

4- Several of the maps remain insufficiently informative. In particular, contour maps presented in the Appendix of the original submission reveal more clearly the magnetic gradients than the revised figures uploaded on 7 December. In addition, figures using background raster images (Figs. 5, 7, and 9 in version 2) fail to clearly distinguish profiles from meshes, and mix two colour scales, resulting in a confusing visual representation.

5- It is difficult to evaluate distances when using latitude and longitude in the coordinate axis, as is the case in most figures. Better to use meters (m) instead.

6- A diurnal correction is mentioned in the legends of Figures 6 and 8. Which magnetic observatory, or variometer station, was used to this end?

7- The 'Recommendations' presented in Section 3.11 largely reflect common knowledge within the geomagnetic community and are not directly supported by the findings of this study.

8- The 'Conclusions' in Section 4 focus primarily on the general motivation for this type of study, as introduced earlier in the manuscript, while a clearer link to the specific results of the present work would strengthen this section.

SPECIFIC ISSUES: Note that line and figure numbers below refer to version 2 of the manuscript, uploaded on the 7th December.

1- lines 30-31: '...guidelines of the 'Guide for Magnetic Measurements and Observatory Practice' published by the International ...';

2- lines 37-41: Text starting at 'Furthermore, offers of ...' should be removed from the abstract;

3- line 55: '...crust, geomagnetically induced currents in power lines, railways and pipelines, archeological ...";

4- lines 82-83: '...more observatories, improving their spatial distribution across the planet, providing for better instrumentation, with reduced noise levels, improving temporal and amplitude precision. It requires well-trained personnel, continuity of data, robust...'

5- line 100: Delete 'modulus';

6- line 107-112: What about a scalar magnetometer?

7- line 120: '...The site should exhibit minimal spatial gradients.'

8- lines 164-165: In Figure 1, it would improve clarity to color stars differently from triangles.

9- lines 175-176: '...These should not exceed 1nT/m, at the location of the absolute measurement instruments, measured with...'

10- lines 187-188: '... foundations such as the ???'

11- Section 3: For clarity, the numbering of the study sites should be consistent with Table 1. Accordingly, Section 3.1 should refer to sites # 1-4, Section 3.2 to sites # 5-6, Section 3.3.1 to site # 7, Section 3.3.2 to site # 8, and so on;

12- line 239: Which 'issues' were identified?

13- line 245: Reference to Figure 3 appears in the text before Figure 2. This should be

corrected.

**14-** line 246: '... presents a contour plot of the magnetic field intensity F from data measured on...'

**15-** line 247: Elevation contours are not seen in figures 3 or 4. I don't think they are required, but they are mentioned in the text. Why is it expected a correlation between elevation and F contours? We don't see such a correlation in Figures 5, 6, 8, 10...

**16-** line 250: Not clear what is meant by 'bottom of Figure 3';

**17-** lines 259-260: The choice of the exterior site to make a profile seems to have been made based on where internal and external reference points coincide. Please explain the importance of the measurements represented in Figure 2. It is missing information on the magnetic measurements along the mine axis.

**18-** line 270: Does Figure 3 represent mesh # 3 in Table 2? Please clarify in the legend. Does Figure 4 represent mesh #7? Needs also to be made clear. In Legend of Figure 3, 'Elevated' should be 'Highest'. Note that the 'background' value removed is different in Figures 3 and 4. Is this a mistake?

**19-** line 282: It doesn't seem necessary to add Table 3, since it has one single row.

**20-** lines 284-286: Not clear why this site was discarded. Needs a more careful explanation, based on the magnetic measurements that were obtained. What was the altitude of the survey?

**21-** line 313: Correct to '(Table 4: survey Figures 5, 6(a) and 6(b)).'

**22-** line 313: '...anomalies persisted, which were attributed to...'

**23-** line 319: Figure 5: It is expected that aerial magnetic values are lower than ground values. But why are TOTAL mesh values as low as the aerial values? Also, what is the altitude of the aerial survey? Profile 3 is missing. The mesh polygons are not clearly delineated.

**24-** line 320: The elevation contours in figures 6a-f, make it difficult to identify clearly the magnetic anomalies. Better figures can be found in the Annexes of the first submitted version of the manuscript (figures 14, 16, 17, 18). Note that although Figures 6b–d cover overlapping regions, the magnetic contours differ significantly. Also, Figure 6f) is the representation of a mesh of points, although it is labeled as representing profiles 11, 12, and 13.

**25-** lines 342-346: Why is the path of the shown profiles so irregular, and why do magnetic measurements oscillate so much? It is missing a discussion on the obtained results.

**26-** lines 355-356: Shouldn't it be Figures 9 instead of Figures 18?

**27-** line 358: What does it mean 'daily prospecting contour plots'?

**28-** lines 360-362: Figure 9 is missing the location of the best site to install the observatory; profile 19 is also missing; profiles 10a and 10b are missing.

**29-** lines 362-364: Figure 10: These figures are very rough and need to be worked out. Profiles # 10, 11, and 12 are represented as meshes; there seems to be some mistake.

**30-** Section 3.9: How is this information related to the installation of a magnetic observatory?

**31-** line 478: Alexandra Pais did not conceive this project. Please correct.

---

## Referee Comment (RC2)

The manuscript primarily recounts the extensive work the team conducted in Honduras for electromagnetic detection. It reads more like an engineering report than a research paper.

Regarding the content, the following questions and suggestions are raised:

1. The article mentions the team's geomagnetic detection work in Honduras. The obtained geomagnetic data should be analyzed in depth to reveal substantial advantages and compare it with other sites, rather than providing superficial analysis or simply using others' data for explanation.

2. The article devotes considerable space to describing the difficulties and challenges of the work; it is recommended to reduce this section appropriately.

3. What is the magnitude and three components of the geomagnetic intensity at the selected location?

4. The key specifications of the equipment selected during the survey should be listed.

5. If a geomagnetic observatory were to be built at the selected location, what are the plans and arrangements, and what equipment will be deployed?